# Causal discovery with endogenous context variables

**Wiebke Günther**[1,2*]   **Oana-Iuliana Popescu**[1,2*]   **Martin Rabel**[1,3,4*]
**Urmi Ninad**[1,2]   **Andreas Gerhardus**[1]   **Jakob Runge**[1,2,3,4]

[1]German Aerospace Center (DLR), Institute of Data Science, Jena, Germany
[2]Technische Universität Berlin, Institute of Computer Engineering
and Microelectronics, Berlin, Germany
[3]Center for Scalable Data Analytics and Artificial Intelligence (ScaDS.AI)
Dresden / Leipzig, Germany
[4]Technische Universität Dresden, Faculty of Computer Science, Dresden, Germany
`oana-iulia.popescu@dlr.de`
*equal contribution

## Abstract

Causal systems often exhibit variations of the underlying causal mechanisms between the variables of the system. Often, these changes are driven by different environments or internal states in which the system operates, and we refer to context variables as those variables that indicate this change in causal mechanisms. An example are the causal relations in soil moisture-temperature interactions and their dependence on soil moisture regimes: Dry soil triggers a dependence of soil moisture on latent heat, while environments with wet soil do not feature such a feedback, making it a *context-specific* property. Crucially, a regime or context variable such as soil moisture need not be exogenous and can be influenced by the dynamical system variables – precipitation can make a dry soil wet – leading to joint systems with endogenous context variables. In this work we investigate the assumptions for constraint-based causal discovery of context-specific information in systems with endogenous context variables. We show that naive approaches such as learning different regime graphs on masked data, or pooling all data, can lead to uninformative results. We propose an adaptive constraint-based discovery algorithm and give a detailed discussion on the connection to structural causal models, including sufficiency assumptions, which allow to prove the soundness of our algorithm and to interpret the results causally. Numerical experiments demonstrate the performance of the proposed method over alternative baselines, but they also unveil current limitations of our method.

## 1 Introduction

In recent years, causal discovery (CD), that is, learning cause-effect relationships from observational data, has garnered significant interest across various domains, from Earth sciences and genomics to industrial settings (11; 26; 27; 1). Such complex systems often exhibit a change of causal mechanisms over time or across different environments, including changes in the associated causal graph. An example are the causal relations between surface heat and latent heat and their dependence on soil moisture regimes: When the soil is dry, it triggers a dependence of soil moisture on latent heat due to the dry soil reflecting heat, while environments with wet soil do not feature such a feedback. These changing causal mechanisms can be represented using so-called contexts, where the causal mechanisms and structures vary depending on the states of one or more context variables that augment the assumed underlying structural causal model (7; 17; 15). Both time-series (29) and non-time-series data can exhibit such behavior. Context-dependent systems have been widely studied; see Mooij et al.

(15) for a review. Context variables (15; 34; 18) have also been called selection variables (17) or regime indicators (23; 29) in the literature.

Causal discovery on such systems is challenging (7; 15) and methods for CD in case of systems with changing causal mechanisms typically assume that the context variable is exogenous, as will be discussed below and in Sec. 3.2. Yet there exist many real-world examples where the context indicators are endogenous. A regime/context variable such as soil moisture can be influenced by the dynamical system variables – precipitation can make a dry soil wet – leading to joint systems with endogenous context variables, as illustrated in Fig. 1, where we show a hypothesized causal model of the drivers of soil moisture (SM). Understanding context-specific causal relationships is crucial for applications such as extreme event forecasting and for enhancing our general understanding of dynamical systems.

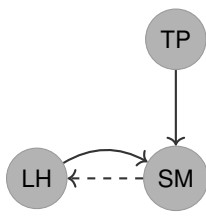

Figure 1: In this strongly simplified example, the variable soil moisture (SM) is an endogenous context variable influenced by variables such as precipitation (TP) and latent heat flux (LH) in both wet and dry regimes of SM, represented by solid edges. In dry conditions only, a feedback loop from the latent heat flux to the soil moisture is created because the soil additionally reflects the heat, represented by the dashed edge (a feedback loop is a time-delayed cycle, leading to a cyclic summary graph representation while the underlying time-series graph remains acyclic).

Our work investigates the assumptions for constraint-based CD on such systems. A core difficulty in treating endogenous context indicators is that context-specific CD methods which resort to masking, i. e., conditioning on the context of interest and ignoring data from other contexts, e.g., (7; 29), can introduce selection bias if the assumption of exogenous context variables is violated. For example, applying CD only on samples from the dry context in the soil moisture problem of Fig. 1 would introduce a spurious correlation between the precipitation and latent heat flux. Other methods apply CD on the pooled data, i.e., the concatenated dataset containing all contexts. Then allowing for endogenous variables is possible (15), but does not yield context-specific information, i.e., no distinction between solid and dashed edges in Fig. 1 can be made. We propose an adaptive constraint-based discovery algorithm that allows to distinguish context-specific causal graphs and give a detailed discussion of the connection to structural causal models, including sufficiency assumptions, which allow to prove the soundness of our algorithm and to interpret the results causally.

Context-specific graphs contain more information than union graphs on pooled data, i.e., the dashed edges in in Fig. 1. This information can under suitable assumptions be discovered by additionally taking context-specific independencies (CSIs) into account. CSIs are independence relations that are only present when data of a single context is viewed in isolation. Naively searching through all CSIs would lead to an inflated search space of available independence tests. Further CSIs necessarily use only a reduced sample size which can also increase error rates. Indeed, a simple baseline method we compare against suffers from an increased error rate, whereas our proposed method is able to avoid an inflation of required tests. This is achieved by an easy-to-implement decision rule which allows to adaptively decide for one and only one test to be executed per separating set. Whenever consistent, tests are further executed on the pooled dataset, thus on all available data.

We now summarize our contributions: (1) We propose a general framework for adapting existing constraint-based CD algorithms to extract context-specific information, in addition to joint information, about skeleton graphs when contexts are (potentially) endogenous without inflating the number of required independence tests. (2) We give an SCM-based interpretation of this additional information and unveil interesting and little-studied difficulties whose understanding we consider essential to the problem. (3) We work out the theoretical assumptions under which our proposed method soundly recovers the context-specific skeletons despite these difficulties. (4) We conduct a simulation study to numerically evaluate the performance of the proposed method and understand the failure modes of current approaches.

## 2 Related work

The problem of CD from heterogeneous data sources has gained interest in recent years. Mooij et al. (14) offers a broad overview of the topic and groups recent works on CD for heterogeneous data

into two categories: Works that do CD separately on each context-specific dataset (the data points belonging to a specific context) and works that use the pooled datasets (all data points). Works that obtain context-specific graphs apply CD on the context-specific dataset, where the assignment to contexts is either known (8; 24), identified before applying CD (12; 36), or iteratively learned (29). The information from the context-specific graphs can then be summarized in a single graph, the so-called union graph.

Mooij et al. (14) propose the Joint Causal Inference (JCI) framework to deal with multi-context CD by modeling the context variable and using CD on the pooled datasets to uncover the relationships between the context and system variables, where the context can also be endogenous. Several CD methods can be extended for JCI, e.g., FCI (32). FCI is also applied in (28; 10) on the pooled data to discover causal relationships among all variables. Zhang et al. (39) test for conditional independence between system and context variables on the pooled dataset under the assumption of exogenous context variables. Constraint-based CD for the multi-context case has also been studied specifically for the time-series case, e.g., (29; 33; 5). Among non-constraint-based methods, Peters et al. (19); Heinze-Deml et al. (6) exploit the invariance of the conditional distribution of a target variable given its direct causes across multiple contexts, while Zhou et al. (41) find causal structure by modeling the direct causal effects as functions of the context variables. Although not directly methodically related, CD for cyclical graphs is also relevant to our problem, as cycles can appear between the different contexts (8; 3). Most above-mentioned methods using both context-specific and pooled data assume, either implicitly or explicitly, that the context variable is exogenous. This is possibly due to the fact that, for context-specific approaches, endogenous variables might lead to selection bias, while for methods that use pooled data it suffices to treat endogenous variables as any other variable in the graph. In our work, we discover context-specific causal graphs for the case of endogenous context variables. Assuming access to the context variable, we obtain information from the pooled and context-specific graphs. We focus on the non-time-series case and leave the extension to time-series for future work.

Labeled Directed Acyclic Graphs (LDAGs) (18) are conceptually closest to our work. LDAGs summarize context-specific information in graphical form by assigning a label to each edge according to the context in which dependence arises. Hyttinen et al. (9) propose a constraint-based method to discover LDAGs from data similar to our approach: A variant of the PC algorithm that tests conditional independence of $X \perp\!\!\!\perp Y | R = r$ within each context $R = r$ instead of testing $X \perp\!\!\!\perp Y | R$ on the pooled data. We discuss how our definitions of context-specific graphs fundamentally differ from CSIs in Sec. 3.1 and delimit our work from LDAGs in App. A.

## 3 Formal description

We first briefly outline preliminary definitions (see, e.g., (16) for detailed definitions). We then point out an unexpected difficulty of our approach in the connection to SCMs and its remedy.

**Preliminaries** We work within the framework of structural causal models (SCM) (16). We define the set of observed **endogenous variables** $\{X_i\}_{i \in I}$, which take values in $\mathcal{X}_i$, with finite index set $I$. Let $V := \{X_i \mid i \in I\}$ represent the set of all endogenous variables. The set of (hidden) **exogenous noises** $\{\eta_i\}_{i \in I}$, taking values in $\mathcal{N}_i$ is denoted $U := \{\eta_i \mid i \in I\}$. For any subset $A \subseteq V$, let $\mathcal{X}_A := \prod_{j \in A} \mathcal{X}_j$. We denote $\mathcal{X} := \mathcal{X}_V$ and $\mathcal{N} := \mathcal{N}_U$.

A set of **structural equations** $\mathcal{F} := \{f_i | i \in I\}$ is a set of parent sets $\{\text{Pa}(i) \subseteq V | i \in I\}$ together with functions $f_i : \mathcal{X}_{\text{Pa}(i)} \times \mathcal{N}_i \to \mathcal{X}_i$. An **intervention** $\mathcal{F}_{\text{do}(A=g)}$ on a subset $A \subseteq V$ is a replacement of $f_j \mapsto g_j$ for $j \in A$. We will only need **hard interventions**, $g_j \equiv x_j = \text{const}$ below.

Given a distribution $P_\eta$ of the noises $U = \{\eta_i\}_{i \in I}$, a collection of random variables $V = \{X_i\}_{i \in I}$ **satisfies the structural equations** $\mathcal{F}$ on $U$ if $X_i = f_i((X_j)_{j \in \text{Pa}(i)}, \eta_i)$. An **SCM** $M$ is a triple $M = (V, U, \mathcal{F})$, where $V$ solves the structural equations $\mathcal{F}$ given $U$. The **intervened model** $M_{\text{do}(A=g)}$ is the[1] SCM $M_{\text{do}(A=g)} := (V', U, \mathcal{F}_{\text{do}(A=g)})$. A **causal graph** describes qualitative relations between variables; given a collection of parent-sets $\{\text{Pa}_i\}_{i \in I}$, on the nodes $I$ add a directed edge $p \to i$ for all $i \in I$ and all $p \in \text{Pa}_i$. See next section Sec. 3.1.

---

[1] We assume all SCMs are uniquely solvable, i.e. given $\mathcal{F}$ and $U$, there is a unique $V$ solving these structural equations. This is not a restriction for the per-regime acyclic models considered later.

We observe data from a system with multiple, slightly different SCMs, corresponding to **contexts** and indicated by a categorical **context indicator / variable**, as illustrated in example 3.1. This variable $R \in V$ is a variable whose value indicates the context, i.e., the dataset to which the data point belongs. Therefore, the value-space of $R$ defines all datasets, and its values associate data points to datasets.

**Example 3.1.** Given a binary context indicator variable $R$ and a *multivariate* mechanism of the form

$$Y := f_Y(X, R, \eta_Y) = \mathbb{1}(R)g(X) + \eta_Y,$$

the dependence $X \to Y$ is present in the context $R = 1$, but absent for $R = 0$. This entails a **context-specific independence** (CSI) $X \perp\!\!\!\perp Y | R = 0$ and $X \not\perp\!\!\!\perp Y | R = 1$.

## 3.1 SCMs for endogenous contexts and their associated graphs

In order to create a connection between CSIs and SCMs, we define multiple graphical objects. We motivate their necessity and exemplify them in Sec. 3.2.

Given a set of structural equations, $\mathcal{F}$, we define the **mechanism graph** $G[\mathcal{F}]$, constructed via parent sets $\text{Pa}_i$ specified via $\mathcal{F}$ (see "causal graphs" above). Given a pair consisting of a set of structural equations $\mathcal{F}$, and the (support of) a distribution of the observables $P(V)$ we define the **observable graph** $G[\mathcal{F}, P]$, constructed via parent sets $\text{Pa}_i' \subseteq \text{Pa}_i$ (of $X_i$), such that $j \notin \text{Pa}_i'$ if and only if for all values $\bar{\text{pa}}$ of $\text{Pa}_i - \{j\}$ the mapping $x_j \mapsto f_i|_{\text{supp } P(\text{Pa}_i)}(x_j, \bar{\text{pa}}, \eta_i)$ is constant (where defined).

The **union graph** of an SCM $M = (V^M, U^M, \mathcal{F}^M)$, where $V^M$ is distributed according to $P_M$, is the observable graph

$$G^{\text{union}}[M] := G[\mathcal{F}^M, P_M]$$

which consolidates the information from multiple contexts into a unified graphical representation. If $\mathcal{F} = \mathcal{F}^{\min}$ satisfies standard minimality assumptions (2, Def. 2.6), then $G[\mathcal{F}^{\min}] = G^{\text{union}}[M]$ (given faithfulness, Sec. 4.2) thus corresponds to the causal graph in the standard sense. Given an SCM $M$ with a context indicator $R$, there are *multiple* meaningful sets of structural equations and *multiple* meaningful associated distributions on observables, which result in *multiple* observable graphs associated to the SCM $M$; an overview is provided in Table 1 in the appendix, see also Fig. 2 and the next subsection. We thus distinguish *multiple* context-specific graphs. The **descriptive graph**

$$\bar{G}_{R=r}^{\text{descr}}[M] := G[\mathcal{F}_{\text{do}(R=r)}^M, P_M(V | R = r)],$$

captures relations present in the *intervened* model's mechanisms $\mathcal{F}_{\text{do}(R=r)}^M$ (the SCM in the context $r$) and encountered in *observational* data together with this context; the last point is formalized by restricting our attention to the support of $P_M(V | R = r)$. For edges not involving $R$, the same information is contained in $G[\mathcal{F}^M, P_M(V | R = r)]$ using the unintervened $\mathcal{F}$, which supports the intuition that the context-specific graph captures the overlap of observations and the context of interest. Since $R$ is a constant per-context, we add edges involving $R$ from the union graph to $\bar{G}_{R=r}^{\text{descr}}[M]$ and denote the result $G_{R=r}^{\text{descr}}[M]$. The result is *very different* from an independence graph of the conditional distribution: There are no edges induced by selection bias in $G_{R=r}^{\text{descr}}[M]$.

The **physical graph** encodes the differences from the union graph necessary to describe the context consistently and describes the intuitive notion of $R$ introducing physical changes in mechanisms:

$$G_{R=r}^{\text{phys}} = G[\mathcal{F}_{\text{do}(R=r)}^M, P_M(V)]$$

Importantly, the context-specific graphs defined above *depend only on the SCM*, without making explicit or implicit reference to independence. This sets them apart from graphical representations of independence structure like LDAGs (18): LDAGS are graphical independence models describing context-specific independence (CSI) structure of a dataset. However, a causal analysis requires an understanding of interventional properties, which, in turn, requires inferring knowledge about the causal model properties. In the single-context case, under the faithfulness assumption, knowledge about the causal properties of models is directly connected to the independence structure. As will be explored in Sec. 3.2, this direct connection cannot generally hold in the multi-context case. Thus, an important open problem is the connection of CSI structure to the underlying causal model. Such a relation is given in Sec. 4.2, connecting properties of the underlying causal model represented by context-specific graphs to CSI structure.

### 3.2 Two problems of CD with endogenous context variables

In this section, we describe two main problems encountered with context-specific causal discovery and illustrate them by two examples. In the columns of Fig. 2, we show the different graphs of systems with the following SCMs:

**Example 3.2.** The mechanisms $X := \eta_X, T := \eta_T$ and the binary threshold exceedence $R := [(X + T + \eta_R) > t] \in \{0, 1\}$ agree for both systems. (i) For the system depicted in the left column of Fig. 2, $Y := R \cdot g(T) + \eta_Y$ is a function of $R$ with the dependence on $T$ explicitly disabled for $R = 0$. (ii) On the right-hand side, $Y := \max(T, T_0) + \eta_Y$, is not a function of $R$, but assume $P(T > T_0 | R = 0) < \epsilon$, such that given $R = 0$ automatically $\max(T, T_0) \approx T_0$ and hence the dependence of $Y$ on $T$ becomes negligible for $R = 0$.

**Selection bias** Endogenous context variables can pose a problem for context-specific CD if the context variable is driven by multiple other variables, and is thus a collider. As described in Sec. 2, methods to obtain context-specific graphs typically apply CD on the context-specific dataset, thus essentially conditioning on the context variable. If the context variable is a collider or a descendant thereof, this approach leads to selection bias (red edge in Fig. 2). This is a subtle issue: From an independence structure viewpoint, the dependence between $X$ and $T$ exists yet does not seem to make any intuitive sense and does not admit causal semantics. A formal description in which the absence of this edge is considered correct – as it should be for a causal analysis – requires *the underlying model* as a starting point. Indeed the graphs defined above from the SCM directly (also shown in Fig. 2) consider the absence of this edge correct.

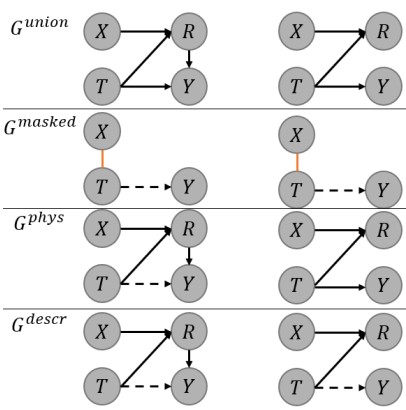

Figure 2: Left: An example of an SCM where the physical and descriptive graphs are the same. Right: an example of an SCM where the physical and descriptive graphs differ, c.f. the "support problem". For $G^{\mathrm{mask}}$, $R$ is not shown, as it is a constant per dataset, and links with other variables will not be found. Context specific graphs depend on the value $r$ of $R$ and are summarized in a single diagram containing a solid edge for edges that are present in both contexts, and a dashed edge for edges that are present for exactly one of the two contexts.

**The support problem** A second problem is encountered, because CSI can arise in two different ways from a given model: Either by an explicitly disabled dependence as in the first example, or indirectly through the lack of observational support in a specific context. Note that, in Fig. 2, our graphical models capture this difference: Once $G^{\mathrm{descr}} = G^{\mathrm{phys}}$ (left-hand side), while in the other case $G^{\mathrm{descr}} \neq G^{\mathrm{phys}}$ (right-hand side).

Concerning the second example in Ex. 3.2 in particular, we make the following observations: **(a)** There is no evidence for the absence of the link $T \to Y$, e. g. when considering interventional questions. **(b)** The absence of the link *does* correctly describe an independence, e. g. edge orientations, cycle resolution, or function fitting on the respective data. Thus, for these problems, the link $T \to Y$ is not required for a meaningful description in the context $R = 0$. **(c)** $R$ is adjacent to $T$, but we could add an intermediate node $T \to M \to R$ without changing the problem, so adjacency of $R$ to the changing link is not guaranteed. **(d)** We lack observations of another context variable $R'$, which indicates whether $T > T_0$, and would suffice to fully describe the qualitative structure of the mechanism at $Y$. Observations (a) and (b) demonstrate that there is not a single "correct" graph to learn, and the additional flexibility of observable graphs introduced above is necessary. (a) and (b) also make it clear, that it should be carefully specified what *should be learned* for a particular application and what *could be learned* (see Sec. 3.1), and what a specific algorithm *will* learn (see Sec. 4.2). Finally, CD heavily relies on restricting the search space for separating sets to adjacencies, which becomes non-trivial in sight of (c). We will return to (d) momentarily.

In our setup, the context indicator is *given* by the user. The context indicator can either arise naturally from the problem setting, or it can be the result of preprocessing using anomaly or regime detection methods (see also Sec. F.2). Since the indicator is chosen, it becomes important to understand which context indicator makes for a *good choice* in view of the problems illustrated above. A core contribution of our work is to understand which assumptions about the model, including the choice

of context indicator, prevent drawing wrong conclusions. We give suitable assumptions to make the problem tractable and the result interpretable (in terms of SCMs) in Sec. 3.3 based on observation (d).

## 3.3 Assumptions

We now define the assumptions that a proper context indicator must fulfill such that our causal discovery method yields meaningful results. These *regime sufficiency assumptions* are described below and further formalized in Sec. C; the general approach and the naming are inspired by observation (d) made above and its resemblance to causal sufficiency.

**Sufficiency assumptions regarding the context indicator** The support problems in Example 3.2 (ii) occur if the qualitative change in the causal mechanisms does not involve the observed context indicator $R$ directly, cf. (d). Therefore, we define the **strong context-sufficiency** assumption (Def. C.1), which enforces that the qualitative changes in the graph involve $R$ directly as displayed in Example 3.1. We also define **single-graph-sufficiency** (Def. C.2), which enforces that certain context-specific graphs collapse to the same graph. This is a weaker assumption than the strong sufficiency above, see Cor. C.1, but still suffices to proof soundness of our algorithm and even allows for a strong interpretation of results as encoding physical changes; this is thus a technically better, albeit less intuitive and potentially harder to verify directly, assumption.

The reduced search space for CSI, cf. (c) in Sec.3.2, happens already under the assumption of **weak context-sufficiency** (Def. C.3), which formalizes dependence on $R$ in the weak sense of being adjacent in the graph. This is yet a weaker assumption, see Cor. C.2, still ensures useful output from our algorithm (Rmk. D.2), but does not allow for a strong (physical) interpretation of the results anymore. These assumptions mirror the causal sufficiency assumption: They exclude hidden contexts and lead to fewer CITs and a stronger interpretation of results as physical changes. We further discuss the hierarchy of these assumptions in Sec. C.1.

**Additional assumptions** We assume **causal sufficiency** (Def. C.4, no hidden confounders). Furthermore, we require at least **weak (descriptive) context-acyclicity** (Def. C.5), i.e., $G^{\mathrm{descr}}_{R=r}$ is a DAG (acyclic) for all $r$. In practice, we assume **strong (descriptive) context-acyclicity** (Def. C.5), meaning there are further no cycles involving any ancestors of $R$ (including $R$ itself) in the union graph. Efficient discovery with our method requires directed cycles of length $\leq 2$, as discussed in Sec. D.7, Sec. D.8. We assume that $R$ is a categorical variable which takes finitely many values $r$ with $P(R=r) > 0$.

## 4 Causal discovery with endogenous context variables

We propose a method to discover context-specific graphs and their corresponding union graph which extends the PC(-stable) algorithm (31; 4) with a novel adaptive testing strategy. In Sec.4.2 we will connect the output of our method to the graphical objects defined in 3.1. In particular, under suitable sufficiency assumptions the output is precisely $G^{\mathrm{phys}}_{R=r}$. The approach is not specific to PC but could also be used with other constraint-based algorithms.

### 4.1 Method

Formal details are given in Sec. 4.2. First observe that for physical changes (edges in $G^{\mathrm{union}}$ not in $G^{\mathrm{phys}}_{R=r}$) or under weak context-sufficiency (Def. C.3), the only changes in $G^{\mathrm{phys}}_{R=r}$ relative to the union graph occur downstream of the context indicator, more precisely in the adjacencies of its children (lemma C.4). On the other hand, spurious links due to selection bias occur upstream of the context indicator (cf. Sec.3.2). By either of the context-acyclicity assumptions introduced above, these two cases are mutually exclusive. Skeleton discovery has to find a valid separating set $Z$ to remove any edge $X$ to $Y$ which is not in the context-specific $G^{\mathrm{phys}}_{R=r}$. By assuming an appropriate version of faithfulness, the decision between independence testing on the context-specific data or on the pooled data does not matter for all other candidates $S$ for $Z$, as long as we decide correctly for (one) valid $S$. On the other hand, under causal sufficiency a valid $S_0 \subseteq \mathrm{Pa}(Y)$ exists by a suitable Markov-property. Thus $R \in S_0$ implies $R$ is a parent of $Y$ and we are in the first case, thus should test on the context-specific data. By mutual exclusiveness, we can consistently do so without the risk of selection bias. Conversely $R \notin S_0$ implies $R$ is not a parent of $Y$, and there is no context-specific

information to be obtained here; thus testing on the pooled data is consistent (it does not loose information), avoids selection-bias ($X$ and $Y$ *could* be ancestors of $R$) and uses all available data. Thus for the *valid* $S_0$ we will by this decision rule always use the correct test.

---

**Algorithm 1** Adaptive CD for Discovering Context-Specific Graphs with Endogenous Context Variables

---

1: **Input:** Context indicator $R$, Samples for $X_1, ..., X_D, X_R = R$
2: **Output:** Context-specific graphs $G_{R=r}$   $\forall r \in \{1, ..., n_r\}$
3: **for** $r = 1$ to $n_r$ **do**
4:     Initialize $G_{R=r}$ as fully connected graph
5:     **for** $sepset_{size} = 0$ to $D - 1$ **do**
6:         **for** each $j \in \{1, \ldots, D, R\}$ **do**
7:             **for** each $i \in \text{Adj}(X_j)$ and $S \subseteq \text{Adj}(X_j) - \{i\}$ with $\#S = sepset_{size}$ **do**
8:                 Test $X_i \perp\!\!\!\perp X_j | S \setminus R, R = r$ if $R \in S$, else test $X_i \perp\!\!\!\perp X_j | S$ on pooled data
9:                 **if** independence is found **then**
10:                     Remove edge $X_i - X_j$ from $G$
11:     Orient edges as in PC algorithm
12: **return** $\{G_{R=r} \mid r \in \{1, ..., n_r\}\}$

---

**Discovering context-specific graphs** First, we discover $G_{R=r}$ which, under the assumptions of Thm. 1, agrees with both the descriptive and the physical graph. For a fixed context value $R = r$, we adapt the skeleton phase of the standard PC algorithm described in (4) to test on the pooled data whenever $R$ is not in the conditioning set. Whenever $R$ is in the conditioning set, the algorithm tests $X \perp\!\!\!\perp Y | \mathbf{Z}, R = r$ instead of $X \perp\!\!\!\perp Y | \mathbf{Z}, R$, as described in Algorithm 1.

**Obtaining the union graph** To obtain the union graph, we combine the information about the edges in all context-specific graphs as follows: If the edge from $X$ to $Y$ is found in none of the context-specific graphs, then no edge is added to the union graph. If an edge $X \to Y$ was found in any of the context-specific graphs, then the directed edge is added to the union graph. The resulting graph may contain both the edge $X \to Y$ and the edge $X \leftarrow Y$. See also Sec. D.14.

**Scaling properties** We study univariate context indicators, but our approach allows multiple indicators. A discussion on the time complexity of our algorithm for this case can be found in Sec. D.9.

## 4.2   Theoretical results

**Markov properties** The Markov property ensures that *links that should be removed, are removed* from suitable independence relations.

**Lemma 4.1.** *Assume strong context-acyclicity, causal sufficiency and single-graph-sufficiency. If $X$ and $Y$, both $X, Y \neq R$, with $Y \notin \text{Anc}_{R=r}^{\text{descr}}(X)$ (this is not a restriction by context-acyclicity, rather fixes notation), are not adjacent in $G_{R=r}^{\text{descr}}$, then there is $Z \subseteq \text{Adj}_{R=r}^{\text{descr}}(Y)$ such that: If $Y$ is neither part of a directed cycle in the union graph (union cycle) nor a child of $R$ in the union graph (union child), then (a) $X \perp\!\!\!\perp Y | Z$, otherwise (b) $X \perp\!\!\!\perp Y | Z, R = r$.*

**Remark 4.1.** If either $X = R$ or $Y = R$, then there is no per-context test available. In this case, we fall back to the standard result: If $X$ and $Y$ are not adjacent in $\text{ACycl}(G^{\text{union}})$, then there is $Z \subseteq \text{Adj}_{R=r}^{\text{descr}}(Y)$ or $Z \subseteq \text{Adj}_{R=r}^{\text{descr}}(X)$ such that $X \perp\!\!\!\perp Y | Z$ (see Sec. D.4).

The full proof can be found in Sec. D.3. The idea is as follows: Case (a) is relatively standard and follows by a "path-blocking" argument (37). We focus on case (b), proved in three steps:

**(i)** Assuming single-graph-sufficiency, $G_{R=r}^{\text{descr}}$ agrees with the graph that would have been observed if $R$ had been intervened to $r$. This graph is, in a standard way, associated to the intervened SCM. Thus, the usual Markov property holds (20, Prop. 6.31 (p. 105)), and independence in that SCM, see (iii), is reduced to d-separation in the context-specific graph. **(ii)** For the cases excluded by case (a), $\text{Pa}_{R=r}^{\text{descr}}(Y) \cup \{R\}$ is always a valid d-separating set (D.2). **(iii)** The intervened model observed under the noises $\eta_i$ of the original SCM, has a counterfactual distribution (Sec. D.2) and by standard results (16, cor. 7.3.2 (p. 229)), the counterfactual independence $X_r \perp\!\!\!\perp Y_r | Z_r, R_r = r$ implies $X \perp\!\!\!\perp Y | Z, R = r$.

**Faithfulness properties** The faithfulness property ensures that *links that should remain in the graph, do remain in the graph* and are not erroneously deleted due to ill-suited independence relations. A probability distribution $P$ is faithful to a DAG $G$ if independence $X \perp\!\!\!\perp_P Y|Z$ with respect to $P$ implies d-separation $X \perp\!\!\!\perp_G Y|Z$ with respect to $G$. This means that, if $G' \subseteq G$ is (strictly) sparser, then faithfulness to $G'$ is (strictly) weaker than faithfulness to $G$. Now, $G_{R=r}^{\text{descr}} \subseteq G^{\text{union}}$, so "$P_M(\ldots)$ is faithful to $G_{R=r}^{\text{descr}}$" is weaker than the standard assumption "$P_M(\ldots)$ is faithful to $G^{\text{union}}$". Similarly (excluding links involving $R$), $\bar{G}_{R=r}^{\text{descr}}$ is sparser than what one would expect for a graph of the conditional model (there are no edges induced by selection bias $\bar{G}_{R=r}^{\text{descr}}$), so "$P_M(\ldots|R=r)$ is faithful to $\bar{G}_{R=r}^{\text{descr}}$" is also weaker than expected. The hypothesis of the following lemma is thus unconventional but not particularly strong. This lemma is proved in Sec. D.5.

**Lemma 4.2.** *Given $r$, assume both $P_M$ is faithful to $G_{R=r}^{\text{descr}}$ and $P_M(\ldots|R=r)$ is faithful to $\bar{G}_{R=r}^{\text{descr}}$ (we will refer to this condition as $r$-faithfulness, or $R$-faithfulness if it holds for all $r$). Then:*

$$\exists Z \text{ s.t. } \left\{ \begin{array}{c} X \perp\!\!\!\perp Y|Z \quad \text{or} \\ X, Y \neq R \text{ and } X \perp\!\!\!\perp Y|Z, R = r \end{array} \right\} \quad \Rightarrow \quad X \text{ and } Y \text{ are not adjacent in } G_{R=r}^{\text{descr}}.$$

**Soundness of the algorithm** *The proposed algorithm recovers a meaningful graph.* Our algorithm is (descriptively) sound (Rmk. D.2) but would not be complete without the sufficiency assumptions introduced in Sec. 3.3. Moreover, the strong- or single-graph-sufficiency makes our algorithm discover the physical graph soundly.

**Theorem 1.** *Assume causal sufficiency, single-graph-sufficiency, $r$-faithfulness and strong context-acyclicity with minimal union cycles of length $\leq 2$ (edge-flips). In the oracle case (if there are no finite-sample errors during independence testing), algorithm 1 recovers the skeleton of $G_{R=r}^{\text{descr}} = G_{R=r}^{\text{phys}}$.*

The detailed proof is in Sec. D.6. Restricting cycles to edge flips is necessary to avoid the problems discussed in Sec. D.7 and Sec. D.8. Together with the context-acyclicity assumption, it ensures that all nodes involved in union cycles are union children of $R$, and thus all edges pointing at union cycles are tested on context-specific data, avoiding problems with acyclifications (see Rmk. 4.1 and (2)). The proof itself consists of three parts: **(i)** Testing *all* subsets $Z \subseteq \text{Adj}(Y)$ as separating sets finds all missing links by the Lemma 4.1. This follows from context-sufficiency and the restricted form of cycles. **(ii)** The algorithm actually does test all such subsets $Z \subseteq \text{Adj}(Y)$. Identical to the argument for the PC algorithm. **(iii)** We do not remove too many edges. This follows directly from Lemma 4.2.

# 5 Simulation study

## 5.1 Experimental setup

**Data generation** Our data models are created by first randomly generating a linear acyclic SCM with up to $D+1$ nodes (base graph) of the form $X^i = \sum_j^{D+1} a_{ij} X^j + c_j \eta^j$, where $i = 1, \ldots, D+1$, at a desired sparsity level $s$. One of the nodes is randomly selected as the context indicator $R$. To obtain categorical context indicators for the $n_{\text{contexts}} - 1$ context-specific SCMs, thresholding is applied by taking the $n_{\text{contexts}} - 1$ quantiles of the continuous values of $R$, adjusted using a data imbalance factor $b$. To generate a context-specific SCM, $n_{\text{change}}$ links of the base graphs are edited sequentially by selecting a variable randomly (excluding the context indicator), adding, removing, or flipping an edge to this variable and ensuring it is a child of $R$ by adding an edge from $R$ to it (and in the case of edge flips, also to the node at the other end of the edge). These operations lead to a non-linear relationship between the context indicator and the other variables. We enforce the presence or absence of cycles in the union graph. The data generation approach is detailed in Sec. E.1.

**Evaluated methods** We compare our method, named PC-AC (AC is short from "adaptive context"), to a variant where PC-stable is applied on each context-specific dataset separately by masking, denoted by PC-M. We expect PC-M to have a higher false positive rate (FPR) due to selection bias, and a lower true positive rate (TPR), as links involving the context indicator cannot be found using the masked data. We also compare our algorithm to PC-stable applied on the pooled dataset, named PC-P, which only recovers joint information, i.e., union graphs. We expect PC-P to have a slightly higher TPR, as it tests all links on all data points. PC-P should not suffer from selection bias, thus, the FPR of PC-P should be lower than that of PC-M for acyclic union graphs. For cyclic union graphs, PC-P only finds the acyclification, thus, the FPR should asymptotically remain finite. We also compare to a

baseline method, named PC-B: For each context, we compute the intersection of the links found by PC-M and PC-P and then add the links found using PC-P for $R$. PC-B should converge asymptotically toward the true graph (see Sec. D.10). However, as PC-B runs more tests overall, PC-AC should have better finite-sample properties. As we do not control for cycle length, the performance of PC-AC might be affected by large cycles, as discussed in Sec. D.7 and Sec. D.8. We obtain union graphs for PC-M and PC-B using the unionization approach from Sec. 4. We also compare union graphs from two methods which test on pooled data: FCI-JCI0 (32; 15) and CD-NOD (39). We do not compare to other methods that obtain context-specific graphs, as they assume exogenous context indicators. A fair comparison to our algorithm is hard to realize due to our assumptions and a lack of available methods suitable to this problem. We compare computational runtimes in App. E.4.

We present results where $D + 1 = 8$ and density $s = 0.4$. For all $j$, $\eta^j \sim \mathcal{N}(0, 1)$. We draw the non-zero $a_{i,j}$ randomly from $\{1.8, 1.5, 1.2, -1.2, -1.5, -1.8\}$, and set $c_j = 1$ for all $j$. For the context indicator $R$, we set $c_R = 0.2$. We discuss results with and without enforced cycles. For the configurations without cycles, we apply only the *remove* operation (see Sec. E.1). The dataset presented here is balanced in the number of samples for each $r$, i.e., $b = 1$. A challenge of our experimental setup is that non-linear relationships between continuous variables and the categorical $R$ must be tested (see also App D.11). Non-parametric permutation-based tests for mixed-type data with non-linear relationships exist (13; 38; 21), but are computationally expensive. Therefore, we apply a parametric mixed-type data test (35) for most experiments. For a fair comparison, we run versions of the algorithms with so-called link assumptions, where the adjacencies of $R$ are either partially or fully known, i.e., we input the (directed) links to or from $R$ (adj. $R$-all), or the (directed) links from $R$ to its children (adj. $R$-ch.), or test without known links (adj. none). For all tests involving only continuous variables, we use partial correlation. When testing pooled data, the link from a regressed-out variable might be missing in one of the contexts, but partial correlation tests should still work, as discussed in Sec. D.12. We show results for "adj. $R$-ch." and "adj. none" and postpone results with "adj. $R$-all" to Sec. E.3. Here, we discuss the TPR and FPR for the union and context-specific graphs (averaged across contexts) but also evaluate edgemark precision and recall in Sec. E.3. Error bars are generated using Bootstrap with 200 iterations. We repeat each experiment 100 times, however, the graph generation is not always successful. We evaluate how well our method scales, e.g., with different $D$, $s$ and $b$ in Sec. E.3. The code, based on causal-learn (40) and Tigramite (25), and details for replication (random seeds) can be found here: https://github.com/oanaipopescu/adaptive_endogenous_contexts.

## 5.2 Results

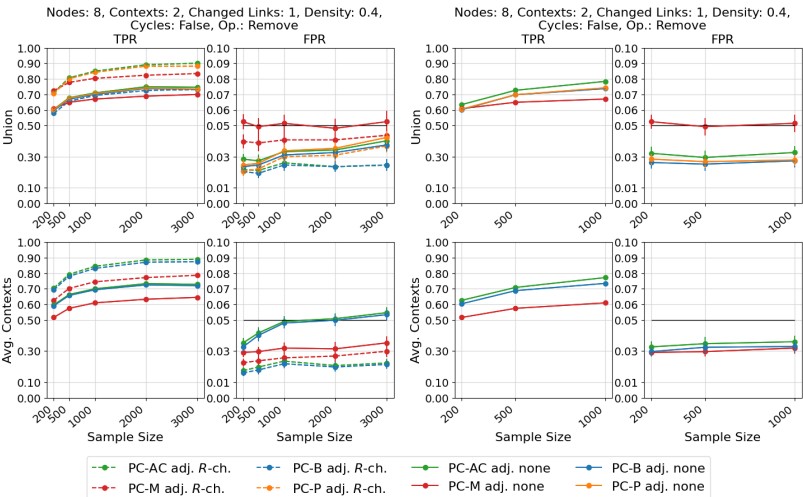

Figure 3: Results for our algorithm PC-AC (adaptive context, our method), PC-M (masking), PC-B (baseline), and PC-P (pooled) on the setup described in Sec. 5.1. Dotted lines indicate settings where links from the context indicator $R$ to its children are fixed/known (adj. $R$-ch.), solid lines indicate no prior knowledge about links (adj. none). The left panel shows results using a parametric CIT, the right panel shows results for the same setting with "adj. none" using a non-parametric CIT.

The left plot of Fig. 3 shows results for the experimental setup described in Sec. 5.1 **without cycles**. For individual contexts, our method outperforms PC-M in terms of FPR for both link assumption cases. Without any link assumptions, the FPR for both PC-B and PC-AC increase significantly. We test whether this is due to the testing problem (Sec. D.11) by using a non-parametric CIT (21). Results in the right plot of Fig. 3 show that the FPR is significantly lowered for all methods. For PC-M, the union graph FPR is high compared to the context-specific FPR, as, for the individual context graphs, the links between context indicator and system variables cannot be found (as $R$ is a constant). Still, false positives can appear in the union graph due to selection bias. Our method performs best in terms of TPR and FPR on union and context-specific graphs with context children link assumptions and without link assumptions. PC-B performs slightly worse due to its finite-sample properties. However, with fully known context-system links (Fig. 4 in Sec. E.3.1), masking finds more links overall, having the best TPR, but also higher FPR, followed by PC-AC, PC-B, and PC-P. Results for the same setup as in Sec. 5.1 **with enforced cycles** in the union graph (Fig. 6 in Sec. E.3.2) show similar behavior of the TPR and FPR, albeit some differences are less pronounced, possibly since the complexity of the problem increases. As expected, the FPR for PC-P, which detects only the acyclification, increases. We observe that for all versions without link assumptions, the FPR increases significantly compared to the setup without cycles. Larger differences in FPR between PC-M and PC-AC can be seen for samples up to 2000. Regarding TPR, PC-AC and PC-B perform best for the context-specific graphs, especially without any link assumptions. Results for the comparison with FCI and CD-NOD for the **union graphs**, detailed in Sec. E.3.3, show that our method slightly outperforms CD-NOD, which assumes acyclicity. Our method outperforms FCI in terms of TPR, but FPR is lower for FCI.

# 6   Discussion and conclusion

In this work, we have introduced a novel method for causal discovery (CD) of context-specific graphs and their union in the presence of a (possibly) endogenous context indicator. We propose a simple modification of the PC algorithm, but our method can be combined with any other constraint-based CD algorithm. Our approach provides an elegant and efficient solution to the problem of selection bias. Further, we discuss how to formally link context-specific independence (CSI) relations to structural causal models (SCMs); to this end we introduce novel graphical objects and assumptions and prove the soundness of our algorithm. We discuss the theoretical and practical limitations to recovering the true graphs, e. g. in presence of large cycles (D.7). Our method has a lower FPR than applying CD on each context individually, which can suffer from selection bias; it displays better finite-sample performance than a proposed baseline method. Our method is comparable with CD-NOD (39) and FCI (28) on the union graphs, but additionally provides context-specific information.

**Limitations** Our method requires that the context indicator $R$ is observed and chosen appropriately, as discussed in Sec. 3.2; it can handle some union cycles, but not reliably so for large union cycles, see Sec. D.8. Without prior knowledge about adjacencies of $R$, our method has difficulties to significantly outperform pooled methods and to keep up with the baseline in numerical tests; this is likely due to the CIT problem discussed in Sec. 5.2, an interpretation supported by a smaller experiment with a non-parametric CIT (Fig. 3, right panel). We thus believe that the problem can be solved using appropriate CITs. Finite sample errors incur further propagated errors similar to standard CD; errors involving links to $R$ additionally lead to a regression to standard CD concerning context-specific information (see App D.13). In this work, we have assumed causal sufficiency, however, real-world data often contains hidden confounders.

**Future work** We have mainly focused on skeleton discovery and omitted a detailed study of orientation rules. Algorithms such as FCI and CD-NOD already perform well for edge orientations of the union graph, as small experiments in Sec. E.3.3 indicate. But there also seem to be additional edge orientations available only by combining per-context information (see Sec. F.1 for an example), a possibility also mentioned in (9); these orientations can be beyond what can be concluded even with JCI arguments. In practice, contexts are often assigned by thresholding, so an extension to contexts given as deterministic functions of observed variables (see Sec. F.2)is of interest. Similar remarks apply to anomaly-detection outputs, defining normal and anomalous contexts. The assumption of causal sufficiency simplifies the arguments, but does not seem strictly necessary; it likely can be relaxed in future work. Finally, our method is in principle extendable to the time-series case. Particularly when allowing for contemporaneous relations of the context indicator, selection bias is a non-trivial problem for time-series.

## Acknowledgments and Disclosure of Funding

We thank Rebecca Jean Herman for the fruitful discussions, and Tom Hochsprung for his helpful comments on the draft. J.R. and U.N. have received funding from the European Research Council (ERC) Starting Grant CausalEarth under the European Union's Horizon 2020 research and innovation program (Grant Agreement No. 948112). J.R., O.I.P., and M.R. have received funding from the European Union's Horizon 2020 research and innovation programme under grant agreement No 101003469 (XAIDA). O.I.P has received funding from the German Aerospace Center (DLR). W.G. was supported by the Helmholtz AI project CausalFlood (grant no. ZT-I-PF-5-11). This work was supported by the German Federal Ministry of Education and Research (BMBF, SCADS22B) and the Saxon State Ministry for Science, Culture and Tourism (SMWK) by funding the competence center for Big Data and AI "ScaDS.AI Dresden/Leipzig".

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

# A  Delimitation from LDAGs

LDAGS (18; 9) are conceptually closest to our work. We thus delimit our work from LDAGs as follows: First, we derive a connection from context-specific graphs to SCMs; also for identification from data, only CSIs are insufficient in our case, a combination of CSIs and independence-statements on the pooled data is required. Second, LDAGs have been studied only under the assumption of acyclicity, while we allow for cycles in the union graph. Third, the method of Hyttinen et al. (9) only accepts categorical variables, while we enable continuous and discrete variables. Fourth, our method reduces the number of tests compared to Hyttinen et al. (9), as we do not test each CI relation conditioned on the context. Instead, we adaptively select whether to test on the context-specific or the pooled data. Finally, we distinguish system variables from the context indicator, whereas in LDAGs every variable is treated as a potential context indicator. From a theoretical perspective, this is advantageous, as it increases information content. However, from a practical perspective, the LDAG approach increases the number of required tests.

# B  Context variables and F-variables

As described near the end of Sec.3.2, in our case, the context variable is *chosen* from the set of variables. This choice is primarily guided the regime-sufficiency assumptions. This allows for a lot of flexibility: The context variable can be selected from or introduced into the dataset by expert knowledge or any regime detection algorithm; it simply has to behave like a random variables from the perspective of the CD-algorithm.

Such flexibility may remind the reader of what Pearl (16) calls $\mathcal{F}$-variables (see below) and while the relation is very indirect, there is indeed an interesting perspective via $\mathcal{F}$-variables. $\mathcal{F}$-variables are functional variables (taking values in function spaces) that allow writing any (hard or soft) interventions implicitly: If $f_Y(x)$ is the mechanism at $x$, depending on parents $x$ (possibly multivariate), one can add a parent $\mathcal{F}_Y$ taking a function (e. g. the original $f_Y$) as value, and replace the mechanism at $Y$ by the evaluation map $\mathrm{eval}(f, x) := f(x)$. Since $\mathcal{F}_Y$ can take a value for $f$ that represents an intervention (e. g. a constant function for a hard intervention) this generalizes interventions. Example 3.1 can indeed be regarded as a kind of soft intervention on $Y$.

Context variables as such are an orthogonal concept: Given multiple distinct datasets over the same set of variables, the context injects the information about the specific associations in each dataset to the analysis by appending a pseudo-variable to the dataset. The value of this pseudo-variable typically is simply a dataset index. This context could be functional but in general both contexts and functional nodes co-exist independently of each other.

Now, we are interested in context-specific changes similar to Example 3.1, which (see above) can be represented as soft interventions, thus as $\mathcal{F}$-variables. This means, we can formally describe these as context-variables which are $\mathcal{F}$-variables. This allows a different interpretation of the strong regime-sufficiency assumption (cf. App.C): It demands, that the context variable belongs to a specific

class of functional nodes, which implement a soft intervention removing some of the parents' effects on certain variables.

In comparison to JCI (15), our method reveals not only the position of this context $\mathcal{F}$-variable in the graph; it also reveals some of the internal structure of some functional nodes (those satisfying the reformulated strong sufficiency assumption): It *additionally* uncovers the graphical changes induced by the implemented soft intervention.

Finally, temporal regimes were an important motivation for our study of context-specific behavior. To make the connection to non-stationary time series, these can be formulated using context variables. For the context to be categorical (which we and others assume), the non-stationarity cannot be a drift etc. but must be driven by a regime change, i.e., there must be temporal regions sharing a model or context. By finding the contexts in which the time series is stationary and describing the causal relationships for those stationary parts of the time series, we can describe the non-stationarity system as a multi-context system.

## C  Details on sufficiency conditions

Here, we provide details on the three context-sufficiency assumptions discussed in Sec. 3.3 in the main text. Then, we show that these are indeed hierarchical: Strong context-sufficiency $\Rightarrow$ single-graph-sufficiency $\Rightarrow$ weak context-sufficiency. Finally, we give examples of when the other direction is violated.

First, we quickly summarize the problem to solve and the ideas behind the different conditions: If the observational support given $R = r$ of, say, $X$, is restricted to a region where a mechanism $f_Y(X, \ldots)$ is constant in $X$, then this can make the link from $X$ to $Y$ disappear in independence testing. Excluding CSIs arising like this requires a way of saying what "constant in a region" means.

The single-graph-sufficiency assumption will capture this requirement on a graphical level. This is powerful for proofs, but does not provide much in terms of explanation and might be hard to verify given a specific model. Example 3.1 shows that there clearly is a rather large and intuitive class of models which look like $f_Y(X, R, \ldots) = \mathbb{1}(R)g(X, \ldots) + h(\ldots)$. Clearly, one still has to ensure that $g$ or $h$ are "constant in a region", so that only the indicator function can remove links. The strong sufficiency condition will capture this intuitive notion, however, we note that this is only one possible formalization (see Rmk. C.1).

As indicated above, the strong sufficiency condition requires a formal way of saying a map is *not* constant on any region. One possible way to do so is to require it to be injective (in every argument, after fixing other arguments):

**Definition C.1.** We call an indicator $R$ **strongly context-sufficient** for an SCM $M$, if both

(a) the $f_i$ are continuous, such that:

(i) If $R \notin \mathrm{Pa}(X_i)$ in the mechanism graph, then $f_i$ injective in each argument, after fixing the other arguments to an arbitrary value.

(ii) If $R \in \mathrm{Pa}(X_i)$ in the mechanism graph, then $f_i$ is of the form

$$f_i(X, X', R, \eta_i) = \mathbb{1}(R \in A) \times g_i(X, X', R, \eta_i) + \mathbb{1}(R \notin A) \times g_i'(X, R, \eta_i),$$

with $A \subseteq \mathcal{X}_R$ and $g_i$, $g_i'$ injective in each argument, after fixing the other arguments to an arbitrary value. This could also be generalized to a sum over multiple indicator functions without changing the results.

(b) for every set of variables $V$ not containing $R$, the distributions $P(V)$, $P(V|R = r)$ and $P(V|\mathrm{do}(R = r))$ have continuous densities.

**Remark C.1.** The notion of "non-constant on any region" could be formalized differently, for example through smooth mechanisms and a requirement on gradients. Further, this is not supposed to be particularly general, but rather to be intuitive. More general conditions to ensure that detected changes are physical is single-graph-sufficiency or that results are (descriptively) complete is weak context-sufficiency (see below).

**Example C.1.** A model with continuous densities of noise-distributions, and mechanisms depending on $R$ only through indicator functions and such that for each value $R = r$ the model $M_{\mathrm{do}(R=r)}$ is linear, is strongly regime sufficient. This includes Example 3.1.

**Definition C.2.** We call a context indicator $R$ **single-graph-sufficient** for an SCM $M$, if

$$G_{R=r}^{\text{descr}} = G_{R=r}^{\text{phys}} = G_{R=r}^{\text{CF}}.$$

with the graphs $G_{R=r}^{\text{descr}}, G_{R=r}^{\text{phys}}, G_{R=r}^{\text{CF}}$ defined as in Sec. D.1.

**Remark C.2.** Under this condition, descriptive results can be interpreted as *physical* mechanism changes.

The efficiency of the algorithm proposed in Sec. 4 comes from the restriction that context-specific changes must occur in children of the context $R$. This requires a weaker condition which we formally describe as weak regime-sufficiency. In this case, the difficult and interesting question to ask is: How should the output of the algorithm be interpreted under this assumption? This is discussed in Rmk. D.2, which can be interpreted as a robustness result: Even when violating the context-sufficiency conditions, the output of the algorithm is *descriptively* correct, and, given weak context-sufficiency, it is still complete.

**Definition C.3.** We call a context indicator $R$ **weakly context-sufficient** for an SCM $M$, if

$$R \notin \text{Pa}^{\text{union}}(Y) \quad \Rightarrow \quad \text{Pa}_{R=r}^{\text{descr}}(Y) = \text{Pa}^{\text{union}}(Y).$$

**Remark C.3.** Under this condition, changes may be *descriptive* only, but the results are complete (all edge removals are found), see Rmk. D.2.

## C.1 Hierarchy of assumptions

Here, we show that above-mentioned assumptions are indeed hierarchical: Strong context-sufficiency $\Rightarrow$ single-graph-sufficiency $\Rightarrow$ weak context-sufficiency. We will start from Def. C.1. The underlying idea for requiring (a) is rather direct:

**Lemma C.1.** *If $X \in \text{Pa}(Y)$ in $G[\mathcal{F}]$, and $f_Y \in \mathcal{F}$ is injective in $X$ after fixing all other arguments, then given $S \subseteq \mathcal{X}_{\text{Pa}(Y)}$ with $x \neq x' \in \mathcal{X}_X$ and $\text{pa}^- \in \mathcal{X}_{\text{Pa}(Y)-\{X\}}$ such that $(x, \text{pa}^-), (x', \text{pa}^-) \in S$, then $f_Y|_S$ is non-constant in $X$.*

*Proof.* $f_Y$ is injective in $X$ after fixing its other arguments to $\text{pa}^-$, thus $f_Y|_S$ is injective in $X$ after fixing its other arguments to $\text{pa}^-$, so $(x, \text{pa}^-) \neq (x', \text{pa}^-)$ implies $f_Y|_S(x, \text{pa}^-) \neq f_Y|_S(x', \text{pa}^-)$. □

Thus, as long as the support $S$ is never singular, i.e., as long as $x \neq x'$ as above exists, restricting to $S$ does not change the parent sets. The continuous densities required in (b) are an intuitive possibility to ensure this:

**Lemma C.2.** *If $X \in \text{Pa}(Y)$ in $G[\mathcal{F}]$ and a distribution $Q$ is given with $Q(X)$ and $Q(\text{Pa}(Y))$ possessing continuous densities, then $\exists x \neq x' \in \mathcal{X}_X$ and $\text{pa}^- \in \mathcal{X}_{\text{Pa}(Y)-\{X\}}$ such that $(x, \text{pa}^-), (x', \text{pa}^-) \in \text{supp}(P(\text{Pa}(Y)|R = r))$.*

*Proof.* If $Q(X)$ has a continuous density $q_x$, it is not a point measure, thus there are $x \neq x'$ with $q_x(x) \neq 0$ and $q_x(x') \neq 0$. With $X \in \text{Pa}(Y)$ and $Q(\text{Pa}(Y))$ possessing a continuous density $q_{\text{pa}}$, we know $q_x = \int_{\text{Pa}(Y)-\{X\}} q_{\text{pa}}(\text{pa}^-, -)d\,\text{pa}^-$ is the marginalization, so $0 \neq q_x(x) = \int_{\text{Pa}(Y)-\{X\}} q_{\text{pa}}(\text{pa}^-, x)$, which implies $\exists \text{pa}'$ with $q_{\text{pa}}(\text{pa}^-, x) \neq 0$. By continuity thus $(x, \text{pa}^-), (x', \text{pa}^-) \in \text{supp}(P(\text{Pa}(Y)|R = r))$. □

Requiring strong context-sufficiency ensures that the only non-injectivities come from $R$, such that after fixing $R$ in $\mathcal{F}_{\text{do}(R=r)}$ and ignoring edges involving $R$ (i.e., using the barred $\bar{G}$ versions):

**Lemma C.3.** *If $M = (\mathcal{F}, U, V)$ is strongly context-sufficient and $R = r$ fixed, then the following are equivalent for $X \neq R \neq Y$:*

*(i) $X \in \text{Pa}(Y)$ in $G[\mathcal{F}_{\text{do}(R=r)}]$*

*(ii) $X \in \text{Pa}(Y)$ in $G_{R=r}^{\text{descr}}[M]$*

*(iii) $X \in \text{Pa}(Y)$ in $G_{R=r}^{\text{phys}}[M]$*

*(iv)* $X \in \mathrm{Pa}(Y)$ *in* $G^{\mathrm{CF}}_{R=r}[M]$

**Corollary C.1.** *If $M$ is strongly context-sufficient, then it is single-graph-sufficient.*

*Proof of the Corollary.* By the lemma, $\bar{G}^{\mathrm{descr}}_{R=r} = \bar{G}^{\mathrm{phys}}_{R=r} = \bar{G}^{\mathrm{CF}}_{R=r}$. The unbarred versions insert the same links involving $R$ from the union graph, thus also $G^{\mathrm{descr}}_{R=r} = G^{\mathrm{phys}}_{R=r} = G^{\mathrm{CF}}_{R=r}$. Recall that this is the definition of single-graph-sufficiency. $\qquad\square$

*Proof of the Lemma.* First note that (ii) $\Rightarrow$ (i), (iii) $\Rightarrow$ (i) and (iv) $\Rightarrow$ (i) are trivial (if a restriction of $f_i$ is non-constant in $X$, then $f_i$ is non-constant in $X$).

Further, $f_i$ which do not depend on $R$ are already injective, and $f_i$ which depend on $R$ are of the form $f_i(X, X', R, \eta_i) = \mathbb{1}(R \in A) \times g_i(X, X', R, \eta_i) + \mathbb{1}(R \notin A) \times g'_i(X, R, \eta_i)$, such that the $g_i$, $g'_i$ are injective. Thus, depending on the value $r$ of $R$ we fixed, either $f_i = g_i$ or $f_i = g'_i$ is in $\mathcal{F}_{\mathrm{do}(R=r)}$, both of which are injective. So all $f_i \in \mathcal{F}_{\mathrm{do}(R=r)}$ are injective.

"(i) $\Rightarrow$ (ii)": Apply lemma C.2, to $Q = P(V|R = r)$ (as a distribution over $V$), to get $\exists x \neq x' \in \mathcal{X}_X$ and $\mathrm{pa}^- \in \mathcal{X}_{\mathrm{Pa}(Y)-\{X\}}$ with $(x, \mathrm{pa}^-), (x', \mathrm{pa}^-) \in S := \mathrm{supp}(P(\mathrm{Pa}(Y)|R = r))$. Thus lemma C.1 applies to yield $f_Y|_S$ is non-constant in $X$, thus by definition of $G^{\mathrm{descr}}_{R=r}[M]$, also $X \in \mathrm{Pa}^{\mathrm{descr}}_{R=r}(Y)$.

"(i) $\Rightarrow$ (iii)": As (i) $\Rightarrow$ (ii) above, but use $Q = P(V)$ in lemma C.2.

"(i) $\Rightarrow$ (iv)": As (i) $\Rightarrow$ (ii) above, but use $Q = P(V|\mathrm{do}(R = r))$ in lemma C.2. $\qquad\square$

Next, recall that we had defined $R$ to be **(weakly) context-sufficient**: if $Y$ is a *not* a child of $R$, then $X \notin \mathrm{Pa}^{\mathrm{descr}}_{R=r}(Y) \Rightarrow X \notin \mathrm{Pa}^{\mathrm{union}}(Y)$. The reason why this is implied by single-graph-sufficiency ($G^{\mathrm{descr}}_{R=r} = G^{\mathrm{phys}}_{R=r} = G^{\mathrm{CF}}_{R=r}$) is that physical changes are caused by $R$, and thus only happen in children of $R$:

**Lemma C.4.** *If $Y \neq R$ with $R \notin \mathrm{Pa}^{\mathrm{union}}(Y)$, then*

$$\mathrm{Pa}^{\mathrm{phys}}_{R=r}(Y) = \mathrm{Pa}^{\mathrm{union}}(Y).$$

*Proof.* By definition, $G^{\mathrm{union}}[M] = G[\mathcal{F}, P_M]$ and $G^{\mathrm{phys}}_{R=r}[M] = G[\mathcal{F}_{\mathrm{do}(R=r)}, P_M]$. By definition, $\mathcal{F}$ and $\mathcal{F}_{\mathrm{do}(R=r)}$ differ only in $f_R$ and (by setting $R = r$) for $f_i$ with $R \in \mathrm{Pa}^{\mathrm{union}}(X_i)$. For $Y$, by hypothesis, neither of these two applies, so the same $f_Y$ is in $\mathcal{F}$ and $\mathcal{F}_{\mathrm{do}(R=r)}$. Since both graph definitions further use the same support (that of $P_M$), their parent definitions for $Y$ agree: $\mathrm{Pa}^{\mathrm{phys}}_{R=r}(Y) = \mathrm{Pa}^{\mathrm{union}}(Y)$. $\qquad\square$

**Corollary C.2.** *If $M$ is single-graph-sufficiency (for $R$), then $M$ is weakly context-sufficient.*

*Proof.* This is a reformulation of the statement of the lemma, i.e. it holds by definition of these terms. $\qquad\square$

## C.2 Strictness: Examples violating the "backwards-direction"

**Part (a) of strong context-sufficiency:** The injectivity assumption is e.g. violated in Example 3.2, if $f_Y$ is not injective.

**Part (b) of strong context-sufficiency:** The assumption about distributions seems, especially in light of Lemma C.2 (which is the only implication of this assumption that is needed), relatively weak. However, there are some intuitive examples where we observe that it can be easily violated, and why this is a problem:

**Example C.2.** Assume a root variable $X$ has a noise term, i.e., mechanism kernel $(f_X)_* \eta_X$, with randomness arising (within measurement precision) from a process involving sunlight, e.g., $X$ might be the photosynthesis rate or the solar power provided. Further, assume that $R$ indicates whether it is day or night. Then $P(X|R = r)$ is, for some values of $R$ during the night, singular at a specific value, violating part (b) of the strong sufficiency assumption.

On the pooled data, this may not pose an issue. However, on the night-time values of $R$, nothing will depend on $X$. This is descriptively correct, but it holds for all children $Y$ of $X$, irrespective of

whether $Y$ are also children of $R$. Therefore, $R$ is not even weakly context-sufficient. Our algorithm, which relies on weak context-sufficiency, may not remove these links.

This kind of problem may be less severe than a violation of part (a) of strong context-sufficiency because it may be testable beforehand.

**Single-graph-sufficient, but not strongly context-sufficient:** Part (b), as stated cf. Lemma C.2, however, can be violated simply by constructing densities with "jumps," which does not interfere with any of the other conditions. More interesting seem violations of part (a). However, this can also easily be done because the $f_i$ need not be injective, only "sufficiently injective" to be non-constant on the supports. For example, $Y = X^2 + \eta_Y$ is non-injective but is unlikely to be constant (indeed, this would only happen for $P(X)$ having singular support exactly at two points $x_0$ and $-x_0$, thus violating part (b)).

**Weakly context-sufficient, but not single-graph-sufficient:** Say we take Example 3.2 and add a quantitative dependence of $Y$ on $R$, e. g., add a term $Y = \ldots + \gamma \times R$ with $\gamma \neq 0$ to the definition of $Y$. Now, $R$ is a cause of $Y$, turning $Y$ into a child of the context indicator, making the model weakly context-sufficient. However, $f_Y$ still depends on $T$ in the support of $P_M$, such that $T \in \mathrm{Pa}_{R=r}^{\mathrm{phys}}(Y)$ (as before), while (also as before) $T \notin \mathrm{Pa}_{R=r}^{\mathrm{descr}}(Y)$.

In the intuitive sense, weak context-sufficiency means that descriptive changes only occur at mechanisms that involve $R$, but $R$ may, as in the example above, not directly cause the change of the mechanism.

## C.3  Additional assumptions

**Definition C.4** (Causal Sufficiency). A set of variables $V$ is **causally sufficient** relative to an SCM if, within the model, every common cause of any pair of variables in $V$ is included in $V$, i.e., there are no hidden common causes or confounders outside $V$ that affect any pair of variables in $V$.

**Definition C.5** (Weak (Descriptive) Context-Acyclicity). For each context value $r$ of $R$, the context-specific graph $G_{R=r}^{\mathrm{descr}}$ is **weakly (descriptive) context-acyclic** if $G_{R=r}^{\mathrm{descr}}$ is a DAG, i.e., there are no directed cycles in any $G_{R=r}^{\mathrm{descr}}$.

**Definition C.6** (Strong (Descriptive) Context-Acyclicity). For each context value $r$ of $R$, the context-specific graph $G_{R=r}^{\mathrm{descr}}$ is **strongly (descriptive) context-acyclic** if $G_{R=r}^{\mathrm{descr}}$ is a DAG, i.e., there are no directed cycles in any $G_{R=r}^{\mathrm{descr}}$, and, additionally, no cycles involving any ancestors of $R$, including $R$ itself, are in $G^{\mathrm{union}}$.

# D  Theoretical properties of the proposed algorithm

Here, we give proofs and additional information concerning the theoretical properties of the algorithm proposed in Sec. 4.

## D.1  Additional graphs

As a systematic overview over the graphical objects introduced in Sec. 3.1, these objects are summarized and compared in Table 1.

The proof of the Markov property in Sec. D.3 will require an additional graphical object, the counterfactual graph. As it is only needed as formal objects, it is introduced here rather than in the main text. Furthermore, the proof of the Markov property should be comprehensible without detailed knowledge of these graphical objects beyond the understanding that there is, in general, more than one (cf. points (a) and (b) in Sec. 3.2). We advise the reader to first read Sec. D.3, and refer here for formal details if required.

The **counterfactual graph**, defined as follows:

$$G_{R=r}^{\mathrm{CF}}[M] := G[\mathcal{F}_{\mathrm{do}(R=r)}^M, P_M(V \mid \mathrm{do}(R = r))] = G^{\mathrm{union}}[M_{\mathrm{do}(R=r)}]$$

The counterfactual graph describes relations that we would have observed had $R$ been set to $r$ under the same noises. Refer to Sec. D.2 for a more detailed discussion.

Table 1: Comparison of different observable graphs $G[M, P]$, highlighting their key features and contexts in relation to system and model.

| | Observable Graphs $G[\mathcal{F}, P]$ | | |
|---|---|---|---|
| Symbol | $G_{R=r}^{\text{descr}}$ | $G_{R=r}^{\text{phys}}$ | $G^{\text{union}}$ |
| Name | Descriptive | Physical | Union (Standard) |
| Mechanisms $\mathcal{F}$ | $\mathcal{F}$ or $\mathcal{F}_{\text{do}}$ | $\mathcal{F}_{\text{do}}$ | $\mathcal{F}$ |
| Observational Support $P$ | $P_M(\ldots\|R=r)$ | $P_M(\ldots)$ | $P_M(\ldots)$ |
| Captured Information | Directly Observed | Altered Mechanisms | Union Model |
| Context-Specific ($R$-dependent) | Yes | Yes | No |
| Edge Sets | Active in Context $r \subseteq$ | Present in Context $r \subseteq$ | In Any Context |

**Remark D.1.** This should not be confused with what is sometimes called the "intervened" or "amputated" graph $G_{\text{do}(R)}[M]$; this is obtained from $G^{\text{union}}[M]$ by removing edges pointing into $R$. Generally $G_{\text{do}(R)}[M] \neq G[\mathcal{F}_{\text{do}(R=r)}^M, P_M(V \| \text{do}(R = r))]$, even for edges not involving $R$. This is because the support of $P_M(V \| \text{do}(R = r))$ may be different (potentially even larger) than for $P_M$.

### D.2 Connection to counterfactuals

The definition of the intervened model in Sec. 3 is the SCM $M_{\text{do}(A=g)} := (V', U, \mathcal{F}_{\text{do}(A=g)})$ with $U$ distributed according to $P_\eta$ of $M$. At a per-sample level, i.e., drawing a value $\vec{\eta}$ from $P_\eta$ and running it through both $M$ and $M_{\text{do}(R=r)}$ to obtain $X_i(\vec{\eta})$ and $(X_i)_r(\vec{\eta})$ yields exactly the definition of potential responses given in (16, Def. 7.1.4 (p. 204)). Often, the joint distribution $P(X_i, (X_i)_r)$ (induced from $P_\eta$ by the procedure described above) is of interest. This would mean counterfactually observing both $M$ and $M_{\text{do}(R=r)}$ at the same time. Here we only need $X_i$ or $(X_i)_r$ variables separately, so the "distribution-level" observables (defining $M$ and $M_{\text{do}(R=r)}$ on the same $P_\eta$, but not on the same realizations $\vec{\eta}$) suffice.

From the definition of $X_i(\vec{\eta})$ and $(X_i)_r(\vec{\eta})$ the consistency property defined in (16, cor. 7.3.2 (p. 229)) is relatively easy to see: If, for a particular $\vec{\eta}$, one obtains $R(\vec{\eta}) = r$ from $M$, then solving the equations of $M_{\text{do}(R=r)}$ yields the same result (they only replace the equation for $R(\vec{\eta})$ by $R_r = r$, which for this $\vec{\eta}$ makes no difference). So $P((X_i)_r\|R_r = r) = P(X_i\|R = r)$.

### D.3 Proof of the Markov property

**Lemma D.1.** *Given an SCM $M$ and its visible graph $G = G^{\text{union}}[M]$, conditional $\sigma$-separation (d-separation if $G$ is a DAG) with respect to $G$ implies conditional independence in $P_M$.*

*Proof.* The visible graph $G^{\text{union}}[M]$ is the causal graph in the standard sense, i. e. $G[\mathcal{F}^{\text{min}}]$ with a suitably minimal set of mechanisms (see Sec. 3.1). So the standard results (20, Prop. 6.31 (p. 105)) (for cyclic graphs see e. g. (2) and Sec. D.4) apply to relate $\sigma$-separation (d-separation if $G$ is a DAG) to independence. $\square$

**Lemma 4.1.** *Assuming strong context-acyclicity, causal sufficiency and single-graph-sufficiency. If $X$ and $Y$, both $X, Y \neq R$, with $Y \notin \text{Anc}_{R=r}^{\text{descr}}(X)$ are not adjacent in $G_{R=r}^{\text{descr}}$, then there is $Z \subseteq \text{Adj}_{R=r}^{\text{descr}}(Y)$ such that: If $Y$ is neither part of a directed cycle in the union graph (union cycle) nor a child of $R$ in the union graph (union child), then (a) $X \perp\!\!\!\perp Y\|Z$, otherwise (b) $X \perp\!\!\!\perp Y\|Z, R = r$.*

*Proof of lemma 4.1.* We distinguish the cases corresponding to (a) and (b):

Case (a) [(i) $Y$ is not part of a directed union cycle and (ii) $R$ is not a parent of $Y$]:
Single-graph-sufficiency implies context regime-sufficiency (cor. C.2). $Y \notin \text{Anc}_{R=r}^{\text{descr}}(X)$ (by hypothesis) and (i) $Y$ is not part of a directed union cycle imply $Y \notin \text{Pa}^{\text{union}}(X)$. Thus, if there is a link between $X$ and $Y$ in the union graph, then it is oriented as $X \to Y$. By weak context-sufficiency, and (ii) $R$ is not a parent of $Y$ this is excluded. So $X$ and $Y$ are also not adjacent in the union graph.

Since (i) $Y$ is not part of a directed cycle in $G^{\text{union}}$, a standard path-blocking argument (lemma Sec. D.2) works to show $Z = \text{Pa}^{\text{union}}(Y)$ in $G^{\text{union}}$ $\sigma$-separates $X$ and $Y$. With $G^{\text{union}}[M] =$

$G^{\text{union}}[M]$ being a causal graph in the standard sense (lemma D.1), $\sigma$-separation in $G^{\text{union}}$ implies $X \perp\!\!\!\perp Y | Z$.

Case (b): W. l. o. g., $(*)$ $Y \notin \text{Anc}^{\text{union}}(R)$, otherwise we are in case 1, because if $Y$ were an ancestor of $R$, neither (i) could $Y$ be part of a directed cycle by strong context-acyclicity nor (ii) could $R$ be a parent, thus ancestor, of $Y$, also by strong context-acyclicity.

Define $Z' = \text{Pa}^{\text{descr}}_{R=r}(Y) \cup \{R\}$. Using a standard path-blocking argument, this time in $G^{\text{CF}}_{R=r}$ works: Lemma D.2, is applicable with $Z'$, showing it d-separates $X$ from $Y$, because $Z' \cap \text{Desc}^{\text{union}}(Y) = \emptyset$, as $R$ is not a union-descendant of $Y$ by $(*)$, and $G^{\text{CF}}_{R=r} = G^{\text{descr}}_{R=r}$ is a DAG (by weak context-acyclicity) so parents of $Y$ are also not descendants.

Other than $G^{\text{descr}}_{R=r}$, the counterfactual graph $G^{\text{CF}}_{R=r} = G^{\text{union}}[M_{\text{do}(R=r)}]$ is a causal graph in the standard sense (lemma D.1) of the SCM $M_{\text{do}(R=r)}$. Thus d-separation on the DAG $G^{\text{CF}}_{R=r}$ implies independence $X_r \perp\!\!\!\perp Y_r | Z'_r$, writing $V_r$ for a variable that would have been observed in $M_{\text{do}(R=r)}$ under the same exogenous noises as in $M$. This notation is justified because their distributions are potential responses in the sense of (16, Def. 7.1.4 (p. 204)) (see Sec. D.2).

By definition $R \in Z'$, so we can rewrite this as $Z' = Z \cup \{R\}$ with $R \notin Z$. Now this reads $X_r \perp\!\!\!\perp Y_r | Z_r, R_r$. Generally, $P(X_r | R = r')$ is not identifiable, however, if $r = r'$ it is simply $P(X | R = r)$ by the consistency property of potential responses (16, cor. 7.3.2 (p. 229)) (intuitively, under conditions which lead to $R = r$ anyway, if we had intervened to set $R = r$, that would not have resulted in any changes in the values of the other variables, and therefore, the conditional independencies). Thus

$$X_r \perp\!\!\!\perp Y_r | Z_r, R_r \quad \Rightarrow \quad X_r \perp\!\!\!\perp Y_r | Z_r, R_r = r \quad \Rightarrow \quad X \perp\!\!\!\perp Y | Z, R = r.$$

$\square$

## D.4 Path blocking

We use the definitions of $\sigma$-separation (2, A.16) and the induced Markov property (note that in the reference Markov property refers to $\sigma$-separation implying independence, while our definition refers to non-adjacency implying independence or CSI) (2, A.21). The following path-blocking argument is rather standard (see e. g. (37)), however, we formulate the hypothesis slightly different for ease of application.

**Lemma D.2.** *Given a directed graph $G$ and the non-adjacent (in $G$) nodes $X, Y$ with $Y \notin \text{Anc}_G(X)$, and $Y$ not part of a directed cycle, then any $Z$ with $\text{Pa}_G(Y) \subseteq Z$ and $Z \cap \text{Desc}_G(Y) = \emptyset$ d-separates and $\sigma$-separates $X$ from $Y$ in $G$.*

*Proof.* Let $\gamma$ be an arbitrary path from $X$ to $Y$ in $G$. If the last node $W_n$ along $\gamma$ before reaching $Y$ itself is in $Z$, then $\gamma$ is blocked, because the last edge is then the edge $W_n \to Y$ from a parent $W_n$ to $Y$ pointing at $Y$, such that $W_n$ is not a collider along $\gamma$. Thus $\gamma$ is blocked in the d-separation sense, but $W_n$ also points at $Y$ along $\gamma$ with $Y$ in a different strongly connected component as $W_n$ (because $Y$ is not part of a directed cycle, its strongly connected component is $\text{sc}(Y) = \{Y\}$), so it is also blocked in the $\sigma$-separation sense.

Otherwise (if the last node $W_n \notin Z$), $W_n$ must be a child of $Y$, and $\gamma$ ends as $\gamma = [X \ldots W_n \leftarrow Y]$. Since $Y \notin \text{Anc}_G(X)$, the edges along $\gamma$ cannot all point towards $X$. Let $W_k$ be the node to the right of the last (closest to $Y$) arrow pointing to the right $\gamma = [X \ldots \to W_k \leftarrow \ldots \leftarrow Y]$, then $W_k$ is a collider along $\gamma$.

But the part of $\gamma$ between $W_k$ and $Y$ is a directed path from $Y$ to $W_k$ (because $W_k$ was chosen after the *last* arrow pointing to the right), so $W_k \in \text{Desc}_G(Y)$ and so are all descendants of $W_k$, $\text{Desc}_G(W_k) \subseteq \text{Desc}_G(Y)$. Thus, $\text{Desc}_G(W_k) \cap Z = \emptyset$ and $\gamma$ is blocked both in the d-separation and $\sigma$-separation sense. $\square$

## D.5 Proof of the Faithfulness statement

The hypothesis of the following lemma is motivated in the main text in Sec. 4.2.

**Lemma 4.2.** *Given $r$, assume both $P_M$ is faithful to $G_{R=r}^{\text{descr}}$ and $P_M(\ldots|R=r)$ is faithful to $\bar{G}_{R=r}^{\text{descr}}$. Then:*

$$\exists Z \text{ s.t.} \left\{ \begin{array}{c} X \perp\!\!\!\perp Y | Z \quad \text{or} \\ X, Y \neq R \text{ and } X \perp\!\!\!\perp Y | Z, R = r \end{array} \right\} \quad \Rightarrow \quad X \text{ and } Y \text{ are not adjacent in } G_{R=r}^{\text{descr}}$$

*We will refer to this condition as $r$-faithfulness, or $R$-faithfulness if it holds for all $r$.*

*Proof.* The statement is symmetric under exchange of $X$ and $Y$, so it is enough to show $X \notin \text{Pa}_{R=r}^{\text{descr}}(Y)$. We do so by contradiction: Assume $X \in \text{Pa}_{R=r}^{\text{descr}}(Y)$ and let $Z$ be arbitrary. $Z$ can never block the direct path $X \to Y$, so they are never d-separated $X \not\perp\!\!\!\perp_{G_{R=r}^{\text{descr}}} Y | Z$. By (the contra-position of) the faithfulness assumptions, $X \not\perp\!\!\!\perp_P Y | Z$ and if $X, Y \neq R$ also $X \not\perp\!\!\!\perp_P Y | Z, R = r$ (the second statement is by definition the same as $X \not\perp\!\!\!\perp_{P(\ldots|R=r)} Y | Z$). $\qquad\square$

### D.6 Proof of Soundness of the algorithm

**Thm. 1.** *Assume causal sufficiency, single-graph-sufficiency, $r$-faithfulness and strong context-acyclicity with minimal union cycles of length $\leq 2$ (edge-flips). In the oracle case (if there are no finite-sample errors during independence testing), the algorithm recovers $G_{R=r}^{\text{descr}} = G_{R=r}^{\text{phys}}$.*

*Proof.* (i) We show: If $X$ and $Y$ are not adjacent in $G_{R=r}^{\text{descr}}$, w.l.o.g. by weak context-acyclicity, $Y \notin \text{Anc}_{R=r}^{\text{descr}}(X)$, then an algorithm deletes the edge between $X$ and $Y$ if it tests all subsets $Z \subseteq \text{Adj}_{R=r}^{\text{descr}}(Y)$, on context-specific data if $R \in Z$, on the pooled data otherwise.

Since each $G_{R=r}^{\text{descr}}$ is acyclic, at least one of the edges in a cycle must vanish for each context. Because each edge must appear in at least one context to appear in $G^{\text{union}}$, for a cycle of length $\leq 2$, each directed edge in the cycle must vanish in some context. Thus, already by weak context-sufficiency (implied by the hypothesis, Cor. C.2), all (i.e. both) nodes in the cycle are (union-)children of $R$. Thus, under the restriction to "small" cycles in the hypothesis, part (a) of the Markov property (Lemma 4.1) applies if $R$ is not a parent of $Y$.

If part (a) applies for a $Z$ with $R \in Z$, then writing $Z' := Z - \{R\}$ by $X \perp\!\!\!\perp Y | Z', R \Rightarrow \forall r \; X \perp\!\!\!\perp Y | Z', R = r$, and the tests we do suffice to delete the link. If part (a) does not apply, then (see above) $R$ is a parent of $Y$. Further, at least one of (a) or (b) applies, so (b) applies. With $R$ a parent of (thus adjacent to) $Y$, we test (b).

(ii) We show: The proposed algorithm runs at least all tests required for (i).

The PC-algorithm (and its derivatives like PC-stable or conservative PC) tests for a link $X - Y$ in an efficient way, i.e., *at least* all conditional independencies $X \perp\!\!\!\perp Y | Z$ with $Z \subseteq \text{Pa}_X$ or $Z \subseteq \text{Pa}_Y$ and deletes an edge (starting from the fully connected graph) if this independence cannot be rejected. By the same argument, our proposed algorithm tests *at least* all the conditioning sets required for (i).

(iii) We show: If $X$ and $Y$ are adjacent in $G_{R=r}^{\text{descr}}$, then the edge between $X$ and $Y$ remains in the graph.

By faithfulness (Lemma 4.2), whenever we find a such an independence, the edge is *not* in $G_{R=r}^{\text{descr}}$, i.e. irrespective of which (context-specific) independence tests we actually perform, we never delete an edge that is in $G_{R=r}^{\text{descr}}$. $\qquad\square$

**Remark D.2.** Using the argument of (22, Sec. 4) one can supplement the result of Thm. 1 showing robustness against assumption-violations:
Replacing the assumption of "single-graph-sufficient" by "weakly context-sufficient", the algorithm still recovers all descriptive information $G_{R=r}^{\text{descr}}$.
Irrespective of any context-sufficiency assumptions, the algorithm recovers a graph $G_{R=r}^{\text{detect}}$ with $G_{R=r}^{\text{descr}} \subseteq G_{R=r}^{\text{detect}} \subseteq G_{R=r}^{\text{phys}}$, i.e. all edge removals are descriptively correct, and at least all physically changing links are removed.

*Proof.* The arguments in (i) for showing that we only have to test on context-specific data if $R$ is adjacent still apply (they only use weak context-sufficiency). Further, by weak context-acyclicity together with strong context-acyclicity, links that are in the $G^{\text{union}}$ but not in $G_{R=r}^{\text{descr}}$ are never between

two ancestors of $R$ in the union graph (union-ancestors). So by the Markov property (22, Prop. 4.3), also the remainder of (i) still works. Part (ii) is still as in the standard PC-case. Part (iii) still works, because our faithfulness result (Lemma 4.2) is already sufficiently general. □

## D.7 Large cycles

In this section, we discuss how large cycles, which involve more than two edges, i. e. not arising directly from an edge flip, can lead to problems, as exemplified below. However, this problem may not occur directly as discussed in Sec. D.8).

**Context $R = 0$:** A valid separating set for $X$ and $Y$ is $\{W_5\}$ (the only parent of $Y$), so $X \perp\!\!\!\perp Y | W_5, R = 0$. But $R$ is not adjacent to $Y$ (in the union-graph, see below), so this would not be tested by our algorithm (however, see Sec. D.8 for a discussion): Candidates for separating sets are the subsets of adjacencies, and context-specific tests are only considered if $R$ is in the candidate set.

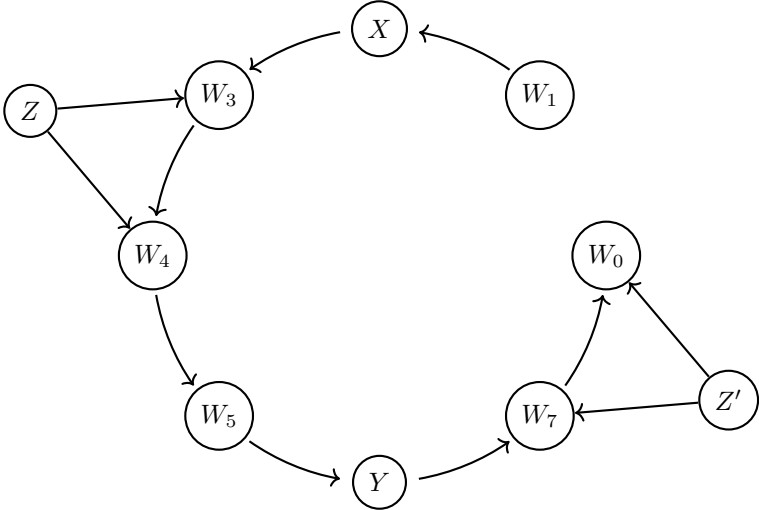

**Context $R = 1$:** A valid separating set for $X$ and $Y$ is $\{W_1\}$ (the only parent of $X$), so $X \perp\!\!\!\perp Y | W_1, R = 1$. But $R$ is not adjacent (see Sec. D.8 however) to $X$, so this would not be tested by our algorithm.

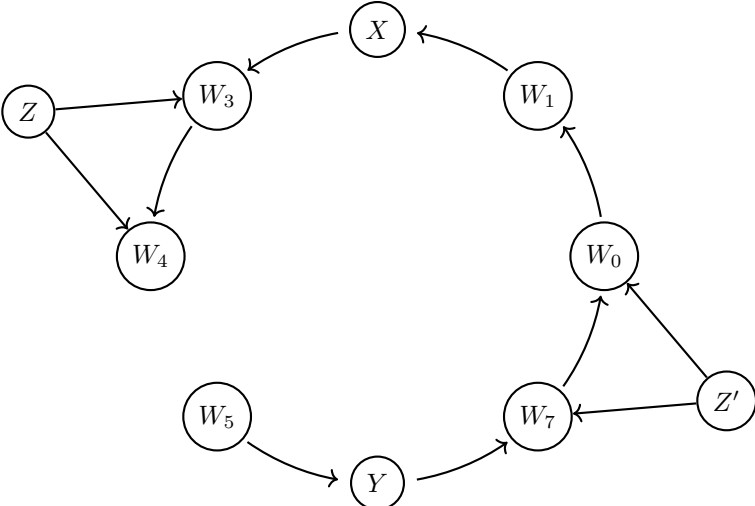

**Ground-truth labeled union graph** Note: $\sigma$-separation implies d-separation. So it is enough to exclude d-separation.

Evidently the empty set is not a separating set $X \not\perp Y$. The parents of either $X$ or $Y$ would only work in one context, in the other they become dependent, so $X \not\perp Y|W_5$ and $X \not\perp Y|W_1$ on the pooled data. For this simple example (without $Z, Z'$), it seems like conditioning on all adjacencies could work. However, this would lead to other problems. For example, with the additional variable $Z$ conditioning on $\mathrm{Adj}(X) = \{W_1, W_3\}$ would open the path $[X \rightarrow W_3 \leftarrow Z \rightarrow W_4 \rightarrow W_5 \rightarrow Y]$ in context $R = 0$. (Similar for $Y$ and $Z'$.)

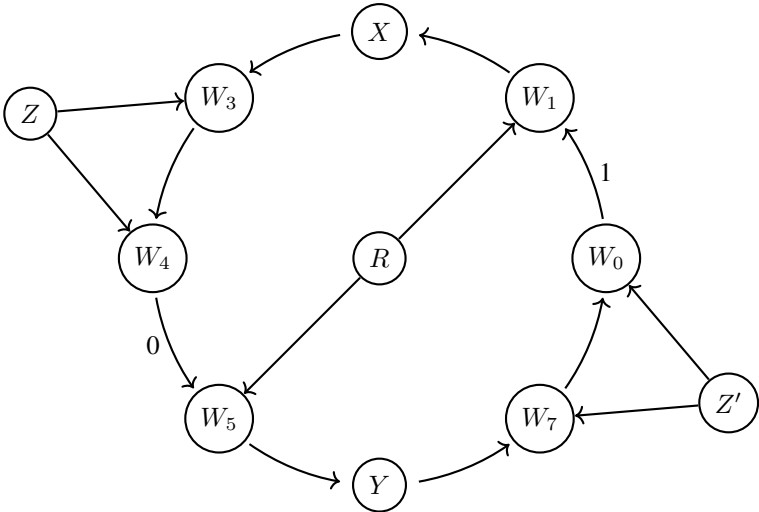

## D.8  Union cycles involving context children

There is another problem with larger union cycles: Links from $R$ to a union cycle cannot be tested using context-specific data (because links involving $R$ cannot be tested for $R = r$, as $R$ is in that case a constant), but tests performed on the pooled data will only delete edges not in the acyclification of the union graph. Thus, if there is a child of $R$ in a union cycle – which is *always* the case given weak context-acyclicity and weak context-sufficiency – then the edges from $R$ to *all* nodes in the cycle cannot be deleted. Thus, $R$ will always point at (be adjacent to, for skeleton discovery) all nodes in a union cycle, irrespective of whether or not they are children of $R$ in the ground truth.

While this is a problem in itself, it seemingly removes the problem of Sec. D.7, so one might be inclined to claim soundness for larger cycles. While this is not entirely wrong, we strongly advise against it, for the following reasons:

(i) Adding *correct* constraints (e. g. from expert knowledge) about missing edges from $R$ to system nodes could break the output.

(ii) In practise (on final data) error-propagation behavior is poor.

(iii) Large cycles can be treated consistently (if they are likely to occur) with the intersection graph method described in Sec. D.10, at the cost of requiring more tests and therefore generally worse finite-sample performance.

It might be possible that (iii) can be used "sparingly": One may observe that the above problem should leave a suspicious trace in discovered graphs: There should be many edges from $R$ to the region of the graph containing the cycle, and all nodes in this region not adjacent to $R$ should be heavily interconnected (because they are not tested on context-specific data, so only the acyclification is found). Thus encountering such a structure may be an indication to test the edges of this region again using the intersection graph method.

## D.9  Complexity

Pairwise tests are always run on the pool in our algorithm, and all context-specific test are "localized" to near the indicator. This means, our algorithm scales like standard PC for increasing variable count but fixed context-variable count. But also for large (or increasing) context-variable count only the possible values of context-variables "nearby" are taken into account as context, leading to an

(approximately) linear scaling: The precise number of tests in each iteration of our algorithm depends on the number of links remaining after the previous iteration and is thus difficult to estimate precisely. However, on finite data, dependencies are typically only detectable for a finite number $H$ of hops along a true causal path. Given that the union graph has a finite graph degree $D_G$, each variable is found dependent (is within $H$ hops) of a finite number $\beta \leq \beta_0(D_G, H)$ of other nodes (upper bounded by a function of graph degree $D_G$ and maximum number of hops $H$ independent of the number of context variables $K$ and the total number of variables $D$, i.e. $\beta_0 = \mathcal{O}_{K,D}(1)$). Every variable is thus after the pairwise tests (pairwise testing is always done on the pool) adjacent to less than $\beta_0$ other variables. We assume[2] that the density $K/D$ of indicators is small compared to $\beta_0$ and each indicator takes at most $C = \mathcal{O}_{K,D}(1)$ many different values. Then on average there are $n \leq \beta_0 \times K/D \times C$ many context-specific tests for each variable. Running up to $n$ CSI tests on average on all $D$ variables leads thus to less than $\beta_0 \times K \times C$ many CSI tests on average, so our algorithm is $\mathcal{O}(K)$ on average. Under the assumption that $K/D$ is fixed as $D$ increases (the density of context variables in the graph is constant), the algorithm is $\mathcal{O}(D^2)$ as is standard PC (the $D^2$ comes from the number of pairwise tests growing faster than the CSI-count).

In comparison, a masking or intersection graph approach, as described in Sec. D.10 has to run all tests for each combination of indicator values, thus scales $\mathcal{O}(\exp(K))$, as each indicator can take at least two values, i.e., there are at least $2^K$ combinations. Thus, our method could scale comparatively well to a larger setups involving many variables and context indicators.

### D.10 Baseline method / intersection graph

From the perspective that information in causal graphs is in the missing links, one would expect that running causal discovery once on the masked / context-specific data (all data points with $R = r$) to obtain $\bar{G}^{\text{mask}}_{R=r}$ (containing edges from selection bias; we exclude $R$ from the graph, as it is a constant) and once on the pooled data (all data points) to obtain $G^{\text{pool}}$ (which in the case of an acyclic union graph is *not* the union graph) one should find all necessary information. Thus, we would expect that the intersection graph obtained by intersecting the obtained sets of edges returns the correct context-specific graphs. Indeed we now also have the theory in place to verify this:

**Lemma D.3.** *The intersection of the edges obtained from the pooled and masked results yield the correct descriptive graph:*

$$\bar{G}^{\text{mask}}_{R=r} \cap \bar{G}^{\text{pool}} = \bar{G}^{\text{descr}}_{R=r}.$$

*Proof.* This follows directly from the Markov property (Lemma 4.1) and faithfulness (Lemma 4.2), even if one only tests $Z$ in adjacencies in the respective discovery algorithms (because both $\text{Adj}^{\text{mask}}_{R=r}(Y) \supseteq \text{Adj}^{\text{descr}}_{R=r}(Y)$ and $\text{Adj}^{\text{pool}}(Y) \supseteq \text{Adj}^{\text{descr}}_{R=r}(Y)$). $\qquad\square$

**Remark D.3.** Generally, the information argument about $\bar{G}^{\text{mask}}_{R=r} \cap \bar{G}^{\text{pool}}$ works as is, but our results show that this has indeed a meaningful connection to the SCM.

This approach even avoids the problems of Sec. D.7. However, it has some severe disadvantages on finite data: Both discovered graphs are individually less sparse than $\bar{G}^{\text{descr}}_{R=r}$, and thus more CITs are necessary (and especially more CITs with large separating sets, which are particularly problematic for both precision and runtime). Furthermore, all tests are executed once on the pooled data for $G^{\text{pool}}$ and once on the masked data $\bar{G}^{\text{mask}}_{R=r}$. Thus, twice as many tests would be necessary, even without the lack of sparsity. With multiple context indicators, the intersection graph approach would scale exponentially while our method scales linearly, as discussed in Sec. D.9.

We use this approach as a baseline to validate the finite-sample properties of our algorithms. We also discuss how to use the intersection graph approach when large cycles may exist in the union graph in Sec. D.8. While large cycles can possibly pose problems for our method, combining our approach with the intersection graph method could yield the correct result in a more efficient way than only using the intersection graph approach.

---

[2]Then within the $\beta$ many adjacencies there is typically at most one indicator; generally we could replace $C$ by $C^{\beta_0}$, as there are at most $\beta_0$ many adjacent indicators with at most $C^{\beta_0}$ many combinations of values, which is still $\mathcal{O}_{K,D}(1)$.

### D.11 Difficulty of detecting context-system links

Testing $R \perp\!\!\!\perp Y$ is not only difficult due to non-linearities, but also due to the following problems: We focus on the case where $R$ is binary for the example $Y = \mathbb{1}(R) \times X + \eta_Y$, and $X = \eta_X$ with both $\eta_X, \eta_Y \sim \mathcal{N}(0, 1)$ standard normal. Here $Y \not\perp\!\!\!\perp R$ (with the empty conditioning set). This is essentially a two-sample test, i. e., we are asking if $P(Y|R = 0) \neq P(Y|R = 1)$. This difference is only visible in variance (higher moments) and not in mean value. $P(Y|R = 0)$ is simply $\eta_Y$ and thus standard-normal $\mathcal{N}(0, 1)$, while $P(Y|R = 1)$ is the sum of two independent standard-normal distributed variables ($\eta_X + \eta_Y$), so it is $\mathcal{N}(0, \sigma^2 = 2)$. Many independence measures like correlation or generalized covariance measure (30) are insensitive to changes in higher moments only (e. g., testing the correlation of $R$ and $Y$ checks for the non-zero slope of the linear fit through $(0, E[Y|R = 0])$ and $(1, E[Y|R = 1])$). Mutual information or distance correlation based tests can gain power against this alternative. Generally, using parametric tests for this scenario (even simple workarounds like testing correlation on squared or absolute values) should also be possible and might provide better power. Some care should be taken (30, Example 1) to formulate the model-class we actually want to test on, as we also have to be able to account for the synergistic (the model is not additive in $X$ and $R$) dependence on $R$.

### D.12 A remark on CIT with regression

Conditional independence testing often relies on regressing out conditions and testing on residuals (e. g. partial correlation). This is usually justified for causal discovery roughly as follows: Only the "true" adjustment set has to work – "erroneous" independence is assumed impossible by faithfulness – and only if $X$ and $Y$ are "truly" non-adjacent. The "true" adjustment-set for $X, Y$ is w. l. o. g. $Z = \mathrm{Pa}(Y)$, with $P(Y|Z) = P(Y|\mathrm{do}(Z))$, so the regression function approximates the structural $f_Y$ which exists and is well-defined as a mapping (in particular single-valued); so for additive noise models testing on residuals should work.

One might, at first glance, consider that this may fail to hold if some links from $Z$ to $Y$ are context-dependent (thus multi-valued). However, this is not the case: If a link from $Z$ to $Y$ vanishes by a change of mechanism, then $R \in Z = \mathrm{Pa}(Y)$ and the *multi-variate* $f_Y$ is *not* multi-valued (on its multi-dimensional input). If a link from $Z$ to $Y$ vanishes by moving out of the observational support, the previous argument still works, i.e., $f_Y$ does not change.

Moreover, if the per-context system (dependencies on variables $\neq R$ after fixing $R = r$) is additive (e. g. linear), then the fit can be done additively because, again, if a link from $Z$ to $Y$ vanishes by a change of mechanism, then $R \in Z = \mathrm{Pa}(Y)$. Since our method tests on data with $R = r$ only, it learns a consistent restriction of the multi-variate $f_Y$ where it is additive. For example, if $\forall r : M_{\mathrm{do}(R=r)}$ is linear with Gaussian noise, then conditional independence between two variables $X, Y$, both $\neq R$, can be tested by partial correlation, and the method remains consistent.

### D.13 A Remark on Error Propagation

Errors involving links to $R$ affect the performance of the algorithm as follows:

For false negatives (a link involving $R$ is not found), no CSI is tested, and thus it does not find additional context specific information, but it also does not incur further errors beyond those inherent to PC.

In the presence of false positives (a link involving $R$ is detected to a non-adjacent $Y$), our method – like PC – executes more tests than strictly necessary, because more adjacencies are available. If $R$ is erroneously found as adjacent, and is added to the conditioning set, in our case the tests will be run as CSI (for $R = r$), i.e., using the context-specific samples only. This can lead to testing on a reduced sample size resulting in further finite sample effects. It should *not* induce selection bias, because we only have to find (at least) one valid separating set, but if $R$ is non-adjacent such a separating set, not containing $R$, exists, still allowing to correctly delete the link.

### D.14 Labeled union graphs

Knowing $G_{R=r}^{\mathrm{descr}}$ for all $r$, we can also construct a labeled union-graph similar to LDAGs (or more precisely – context-specific LDAGs (18, Def. 8)). This follows the philosophy that one obtains strictly

more information than just the union graph, and wants to represent this information as an extension of the union graph rather than in separate graphs.

There are two viewpoints on how to symbolize multiple graphs in one: The information in the graph (about factorization of the distribution) is in the *missing* links, hence, link labels indicate when they "disappear" (stratified graphs and LDAGs follow this convention) – or the edges represent mechanism relationships of the underlying SCM and labels represent when mechanisms are "active." The second option seems more straightforward to read (from a causal model perspective), so we follow this convention below:

- If there is an edge $X \rightarrow Y$ in $G_{R=r}^{\text{descr}}$ for all $r$, insert an (unlabeled) edge $X \rightarrow Y$ into the labeled union graph.

- If there is never an edge $X \rightarrow Y$ in $G_{R=r}^{\text{descr}}$ for any $r$, than there is no edge from $X$ to $Y$ in the labeled union graph.

- If there is an edge $X \rightarrow Y$ for $r \in \mathcal{R}$, and no edge for $r \notin \mathcal{R}$, with $\mathcal{R} \neq \emptyset$ a proper (i.e. not equal to) subset of the values taken by $R$, then add an edge labeled by $\mathcal{R}$, e.g. $X \xrightarrow{\{r_0, r_1, r_5\}} Y$, to the labeled union graph.

# E  Further details on the simulation study

Here, we describe the data generation approach for the numerical experiments in detail. Moreover, we present further results which aim to explore the robustness of our method under different scenarios.

## E.1  Data generation

We generate the simulation data using SCMs. We start by constructing what we will call the base SCM / graph, a linear SCM with up to $D + 1$ nodes, and with acyclic (mechanism) graph, of the form

$$X_i = f_i(\text{Pa}(X_i)) + c_i \eta_i \tag{1}$$

where $i = 1, \ldots, D + 1$, $\text{Pa}(X_i)$ are the parents of $X_i$, $c_j$ are constants, and $\eta_j$ are exogenous noise terms. The function $f_i(\text{Pa}(X_i))$ is defined as:

$$f_i(\text{Pa}(X_i)) = \sum_{X_j \in \text{Pa}(X_i)} a_{ij} X_j. \tag{2}$$

We use a desired sparsity level $s \leq 1$, which leads to the number of total links $L$ (non-zero coefficients $a_{ij}$) to be found as the fraction $s$ of the total possible number of links $D(D + 1)$:

$$L = s \cdot D(D + 1) \tag{3}$$

As a next step, we select a variable to become the context indicator $R$. To this end, we randomly select an index $k \in \{1, ..., D + 1\}$. Then, we assign the variable $X_k$ as the context indicator, i.e., $R = X_k$. For example, if $k = 2$, then $R = X_2$.

As discussed in Sec. 3, $R$ is a *categorical* variable that indicates which of the $n_{\text{contexts}}$ context-specific SCMs the respective sample belongs to. Thus, we must generate categorical values from the continuous values of $X_k$. To achieve this, we define $n_{\text{contexts}} - 1$ quantiles $\{q_1, \ldots, q_{n_{\text{contexts}}-1}\}$ as linearly spaced values between 0 and 1. These quantiles are then adjusted using a balance factor $b$ to compute the final quantiles as $\{q_1^b, \ldots, q_{n_{\text{contexts}}-1}^b\}$. The balance factor $b$ affects the distribution of data points across contexts:

- If $b = 1$, the data points are balanced across contexts.
- If $b < 1$, the balance tips towards the left tail of the distribution.
- If $b > 1$, the balance tips towards the right tail of the distribution.

Using the adjusted quantiles, we compute the values for $R$ as follows:

$$R = \begin{cases} 1 & \text{if } X_k < q_1^b \\ 2 & \text{if } q_1^b \leq X_k < q_2^b \\ \vdots & \vdots \\ n_{\text{contexts}} & \text{if } X_k \geq q_{n_{\text{contexts}}-1}^b \end{cases}$$

Thus, the continuous variable $X_k$ is converted into a categorical variable $R$ indicating the context based on the thresholds defined by the adjusted quantiles. This results in $R$ having a non-linear relationship to all other variables.

Using $R$, we can now proceed to generate context-specific SCMs which have $n_{\text{change}}$ changed links compared to the base graph. For this, we edit the base graph sequentially using the allowed operation set $S_{\text{ops}}$, as follows:

1. Randomly select a variable $X_c$ (excluding $R$),

2. Randomly select an operation from the set $S_{\text{ops}}$,

3. Apply the change to the graph:

    - **Add an Edge:** Add an edge from $X_c$ to another randomly selected variable $X_{c'}$, if an edge between $X_c$ and $X_{c'}$ does not already exist.
    - **Remove an Edge:** Remove an existing edge from $X_c$ to a randomly selected adjacent variable $X_{c'}$.
    - **Flip (Reverse) an Edge:** Reverse an existing edge from $X_c$ to a randomly selected adjacent variable $X_{c'}$, making $X_{c'}$ the parent of $X_c$.

4. Add a link from $R$ to $X_{c'}$. If the selected operation is flip, then add a link from $R$ to $X_c$ and $X_{c'}$.

We note that performing the selected operation on the randomly selected node is not always possible. For example, the node might not have any edges to remove, or adding an edge would lead to the graph being cyclic. We, therefore, check for possible inconsistencies and repeat the process several times (in the case of the experiments presented here, 10 times). If an accepted graph is not obtained after these repetitions, we interrupt the process and continue with a new base graph. Therefore, not all repetitions of an experiment may be successful.

As described in Sec. 5.2, we can either enforce cycles between the context-specific graphs (and thus, cycles in the union graph) or we do not allow them for a specific configuration. If the configuration must contain cycles, for each proposed edit operation, we check whether doing this operation would result in cycles and accept it if so. Otherwise, we reject the operation and start a new trial. This also leads to some repetitions of an experiment not being successful. We mention that we do not enforce a specific cycle length.

Once a context-specific SCM is created, we generate the data according to this SCM. For each variable $X_i$ in the context-specific SCM, we determine its value based on its parents and the structural equations of the SCM. Formally, for a variable $X_i$ with context-specific parents for context $R = r$, $\text{Pa}^{R=r}(X_i)$, its value is generated as:

$$X_i = f_i(\text{Pa}^{R=r}(X_i)) + c_i \eta_i$$

where $f_i$ is defined as in Equation 2. By following this process, we ensure that each variable affected by the context is generated according to the modified SCM, incorporating the context-specific changes introduced during the editing phase.

## E.2   Failed attempts

As mentioned in Sec. 5.2, we repeat each experiment 100 times. However, as discussed above, due to the random data-generating process described above, not all graph generation trials are successful. The accompanying 'failed.txt' file in the .zip file reports how many graph generation trials have failed

for each configuration. Here, we mention that, for the results presented in Sec. 5.2, 10 trials have failed. We also report the number of failed trials in the following subsections which present additional results.

## E.3 Further results

We now present results for further configurations, which vary $D$, $s$ and $b$, as well as detailed discussions of the experiments mentioned in the main paper. All results were obtained by conducting trials in parallel across a cluster of 116 CPU nodes, each equipped with 2x Intel Xeon Platinum 8260 processors. The supplementary .zip file contains the original code used for running the experiments on the cluster. Additionally, we provide an option to run the experiments in sequential mode.

### E.3.1 Further results for the configuration in Sec. 5.2

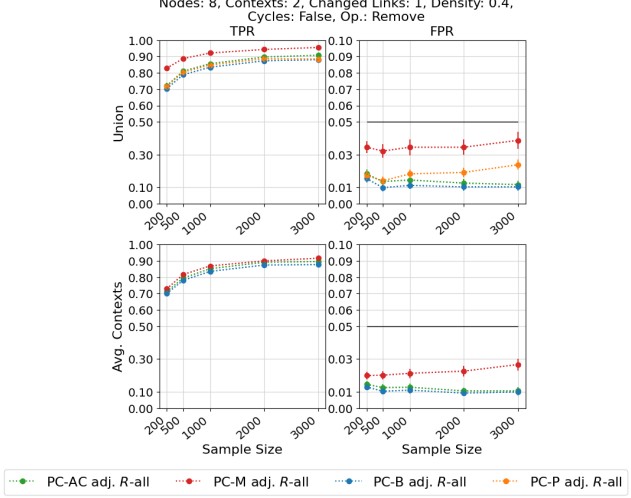

Figure 4: TPR and FPR results for the setup presented in Sec. 5.2 without cycles in the union graph, where all links to $R$ are known (adj.$R$-all) for the methods PC-AC (our algorithm), PC-B (baseline), PC-M (masked data) and PC-P (pooled data). Results for the union graph are presented in the row above, and results for the context-specific graphs averaged over contexts are presented in the row below.

**Results with context-system link assumptions** Fig. 4 presents the results from the setup presented in Sec. 5.1 without cycles where all links to and from $R$ are known. Knowing these links significantly improves the TPR for all methods. Notably, PC-M has the highest TPR and FPR, indicating that only knowing about the context-system links in the endogenous context indicator setting does not suffice for PC-M to discover the correct graphs. The FPR is generally lower for the other methods, indicating fewer false discoveries due to the reduced uncertainty about context-system connections.

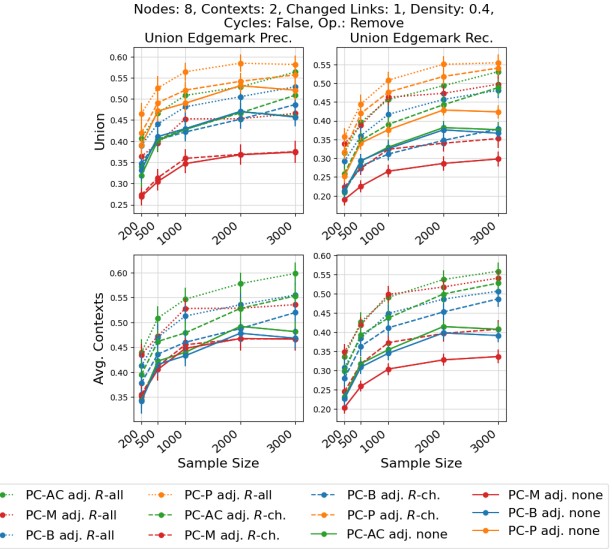

Figure 5: Edgemark precision (prec.) and recall (rec.) results for the setup presented in Sec. 5.2 without cycles in the union graph, where either all links to $R$ are known (adj. $R$-all, dotted line), only the children of $R$ are known (adj. $R$-ch, interrupted line) or no links are known (adj. none, straight line). Here, we present results for the followinf methods: PC-AC (our algorithm), PC-B (baseline), PC-M (PC with masking) and PC-P (PC on pooled data). Results for the union graph are presented in the row above, and results for the context-specific graphs, averaged over contexts, are presented in the row below.

**Edgemark precision and recall**    Fig. 5 shows the precision and recall values for both the union and the context-specific graphs (averaged over contexts) for PC-AC, PC-M, and PC-B and the union graphs for PC-P for the different link assumptions. On the context-specific graphs, we observe that assuming all links to and from the context indicator are found (adj. $R$-all), PC-AC has the lowest precision, i.e., lower risk of falsely oriented edges, and highest recall, i.e., finding the correct orientations, followed PC-B. For the other link assumption cases, our method outperforms all other methods for the context-specific graphs. For the union graph, PC-P scores best, likely because it finds the most links in the union case, followed by our method. PC-M scores last.

### E.3.2    Cycles

Fig. 6 presents results for the experimental setup as described on Sec. 5.2 with the difference that cycles between the context-specific graphs are enforced here. For this experiment, a total of 29 trials have failed (see Sec. E.2). From the total of 71 samples, 69 samples have cycle length 2, and 2 samples have cycle length 3. We observe similar behavior of the TPR and FPR for the non-cyclic results presented in Sec. 5.2. However, some differences are less pronounced, possibly due to the increasing complexity of the problem. As expected, the FPR for PC-P, which only detects the aycyclification, increases. We observe that for all versions without link assumptions, the FPR increases significantly compared to the setup without cycles. The largest differences in FPR between PC-M and PC-AC are for a smaller number of samples, up to 2000. For 3000 samples, the FPR of PC-M is lower than the FPR of PC-AC, possibly due to overall fewer links being tested compared to PC-AC: Since PC-AC also finds context-system links, it tests more adjacent pairs and possibly finds more links. However, in terms of TPR, PC-AC, and PC-B perform best for the context-specific graphs, especially without any link assumptions.

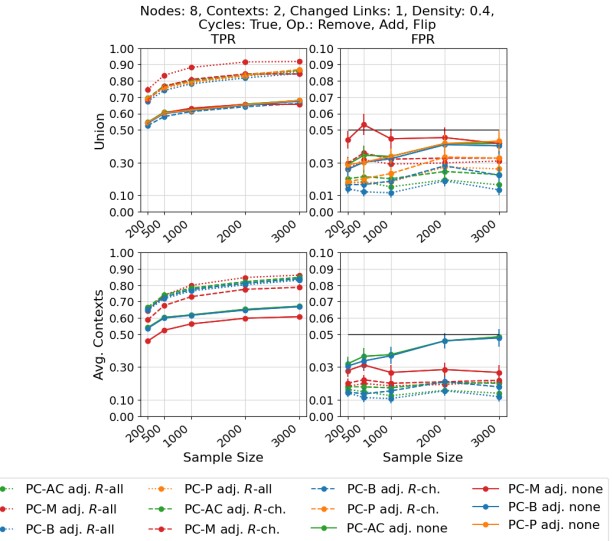

Figure 6: TPR and FPR results for the setup presented in Sec. 5.2 where cycles in the union graph are enforced. Here, we present results where all links to $R$ are known (adj.$R$-all, dotted line), links to children of $R$ are known (adj. $R$-c., interrupted line) or without any link assumptions (adj. none, straight line) for the methods PC-AC (our algorithm), PC-B (baseline), PC-M (PC with masking) and PC-P (PC on pooled data). Results for the union graph are presented in the row above, and results for the context-specific graphs, averaged over contexts, are presented in the row below.

### E.3.3 Comparison with FCI-JCI0 and CD-NOD

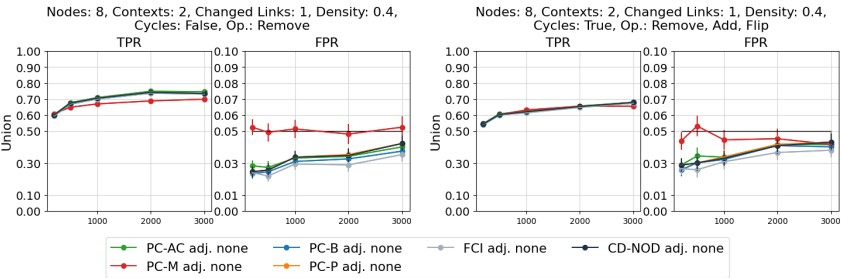

Figure 7: TPR and FPR results for the setup presented in Sec. 5.2 without cycles in the union graph (left plot) and with enforced cycles in the union graph (right plot). Here, we present results where all links to $R$ are known (adj.$R$-all), for the methods PC-AC (our algorithm), PC-B (baseline), PC-M (masked data) and PC-P (pooled data), FCI-JCI0 (FCI) and CD-NOD. Results for the union graph.

**Skeleton discovery**   As mentioned in Sec. 5.2, we compare the results of our algorithm for the discovery of the union graph to the Fast Causal Inference (FCI) algorithm (32), discussed as JCI-FCI0 in Mooij et al. (15). We apply JCI-FCI0, which does not assume exogenous context variables (we name it only FCI from now on) for the multi-context setup, and CD-NOD (39), a causal discovery algorithm for heterogeneous data, which assumes exogenous context variables. Both algorithms are applied to the pooled datasets. We use the causal-learn implementation (40) for both algorithms. Since, for the available implementation, it is not possible to introduce link assumptions from the beginning of the discovery phase (while it is possible to introduce link assumptions, at the moment, they are only used to refine the graph after the discovery phase), we only evaluate the algorithms for the case without link assumptions (adj. none). As the algorithms only use pooled datasets, we do not present any result for the context-specific graphs.

Fig. 8 presents the results for the setup in Sec. 5.2 of the main paper, without cycles (left-most two plots) and with cycles (two right-most plots). We observe that both FCI and CD-NOD behave

similarly to PC-AC. We observe minimal differences between PC-AC and CD-NOD for smaller samples, with our method performing slightly better. FCI performs best across all methods in terms of FPR. Nevertheless, FCI can only recover the union graph, and thus, our method still has the advantage of gaining more descriptive information than FCI. However, our method may perform poorly due to the CIT, and we believe that using an adequate CIT, our method can improve over FCI and CD-NOD.

**Edge orientations**  We observe that FCI and CD-NOD perform best when comparing the precision and recall values for the edge orientations between the different methods. This result is expected: Our method focuses on skeleton discovery and does not propose any special rules to orient edges obtained from the union of the context-specific graphs, which is left for future work, as discussed in Sec. 6. CD-NOD and FCI develop specific orientation rules involving the context indicator, and thus are expected to perform better in terms of orientations at the moment. However, as we consider that there is additional information from the context-specific graphs that can be used to orient more edges, as exemplified in Sec. F.1, we believe that the performance of our method regarding edge orientations can be significantly improved.

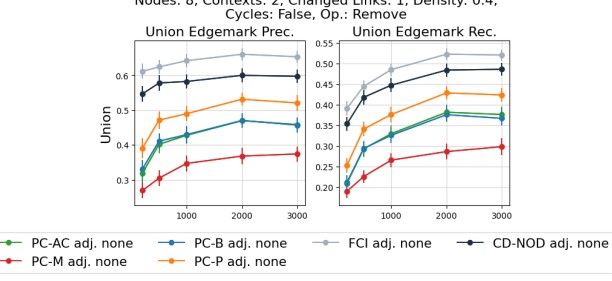

Figure 8: Edgemark precision (prec.) and recall (rec.) results for the setup presented in Sec. 5.2 without cycles in the union graph. Here, we present results where all links to $R$ are known (adj.$R$-all), for the methods PC-AC (our algorithm), PC-B (baseline), PC-M (masked data) and PC-P (pooled data), FCI-JCI0 (FCI) and CD-NOD. Results for the union graph.

### E.3.4 Higher number of changing links

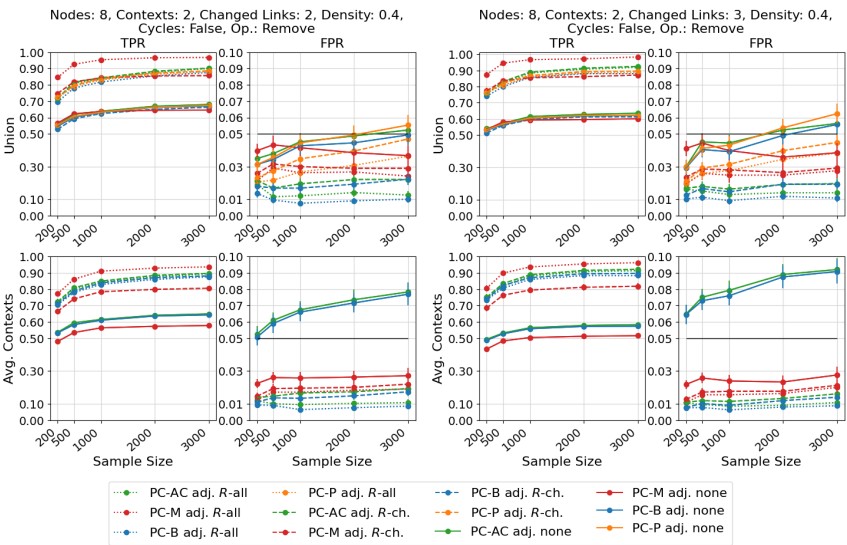

Figure 9: TPR and FPR results for the setup presented in Sec. 5.2 without cycles in the union graph, but with a higher number of changing links $n_{\text{change}} = 2$ (left plot) and $n_{\text{change}} = 3$ (left plot). Here, we present results where all links to $R$ are known (adj.$R$-all), links to children of $R$ are known (adj. $R$-c.) or without any link assumptions (adj. none) for the methods PC-AC (our algorithm), PC-B (baseline), PC-M (masked data) and PC-P (pooled data). Results for the union graph are presented in the rows above, and results for the context-specific graphs averaged over contexts are presented in the rows below.

Here, we explore the scenario where the number of changing links between contexts varies. We keep the experimental setup as in Sec. 5.2, only changing the number of changing links $n_{\text{change}}$ to 2 and 3. Figure 9 shows that as the number of changing links increases, the TPR generally improves, suggesting that the presence of more changes provides more signals that the causal discovery process can detect. Conversely, a higher number of changing links leads to an increase in FPR, especially for the cases without link assumptions. However, we believe this is due to the CIT, which generally leads to a higher FPR, as the results in Sec. 5.2 of the main paper indicate. Our method performs similarly or outperforms PC-P, PC-M, and PC-B for cases with known links from $R$ to its children. We mention that, for this experiment, a total of 18 trials have failed for $n_{\text{change}} = 2$ and 25 trials have failed for $n_{\text{change}} = 3$ (see Sec. E.2).

### E.3.5 Higher graph density

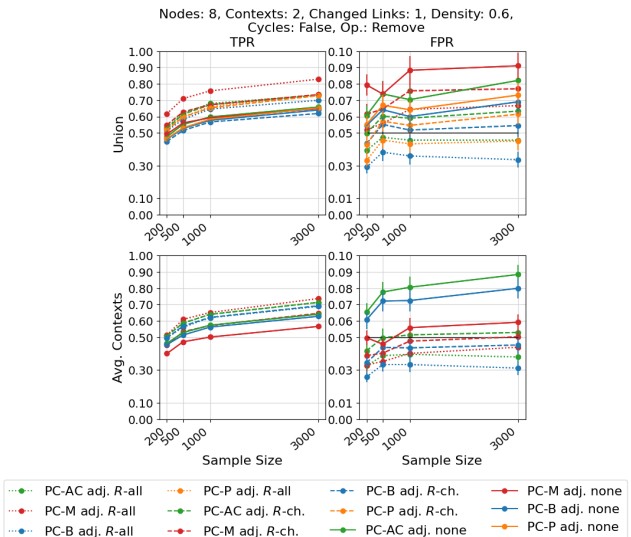

Figure 10: TPR and FPR results for the setup presented in Sec. 5.2, without cycles in the union graph, but with a higher density value $s = 0.6$. Here, we present results where all links to $R$ are known (adj.$R$-all), links to children of $R$ are known (adj. $R$-c.) or without any link assumptions (adj. none) for the methods PC-AC (our algorithm), PC-B (baseline), PC-M (masked data) and PC-P (pooled data). Results for the union graph are presented in the row above, and results for the context-specific graphs averaged over contexts are presented in the row below.

Here, we examine the impact of increasing the density of the base graph on the performance. We keep the experimental setup as in Sec. 5.2, and only the graph density to $s = 0.6$. As Fig. 10 shows, with higher graph density, the TPR decreases slightly compared to the results in the main paper, while the FPR increases. This is likely due to the increased complexity of the problem and possible confounding. The denser the graph, the more candidate links the algorithm needs to evaluate, which can increase the likelihood of falsely identifying spurious links as causal. Our algorithm still outperforms PC-M for known links between $R$ and its children. PC-B scores best overall in terms of FPR, but its finite-sample issues lead to lower TPR results, indicating that it finds fewer links overall. For this experiment, we mention that a total of 12 trials have failed (see Sec. E.2).

### E.3.6 Smaller and larger number of nodes

Fig. 11 present the performance of the algorithms across graphs with different $N = 6$ (left plot) and $N = 12$ (right plot), while the rest of the experimental setup remains as described in Sec. 5.2. Generally, TPR decreases as the number of nodes increases, possibly due to the increased complexity and the higher dimensional space in which the causal discovery process operates. Larger graphs have more potential causal connections, making it harder to distinguish true causal links from noise. The FPR slightly increases with the number of nodes. However, our method (PC-AC) still outperforms PC-M when no links are assumed, or the links from $R$ to its children are known. For this experiment, we mention that a total of 14 trials have failed for $N = 6$ and 12 trials have failed for $N = 12$ (see Sec. E.2).

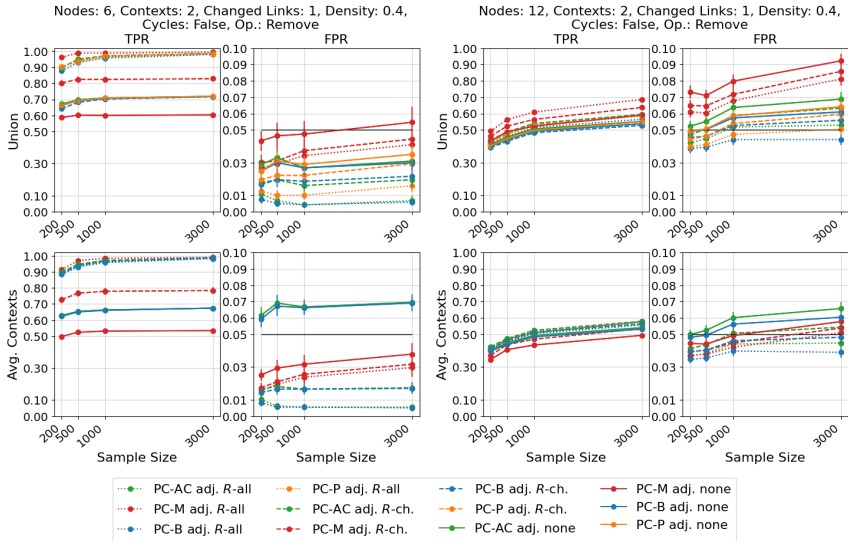

Figure 11: TPR and FPR results for the setup presented in Sec. 5.2 without cycles in the union graph, but with number of nodes $N = 6$ (left plot) and $N = 12$ (right plot). Here, we present results where all links to $R$ are known (adj.$R$-all), links to children of $R$ are known (adj. $R$-c.) or without any link assumptions (adj. none) for the methods PC-AC (our algorithm), PC-B (baseline), PC-M (masked data) and PC-P (pooled data). Results for the union graph are presented in the rows above, and results for the context-specific graphs averaged over contexts are presented in the rows below.

### E.3.7    Data imbalance

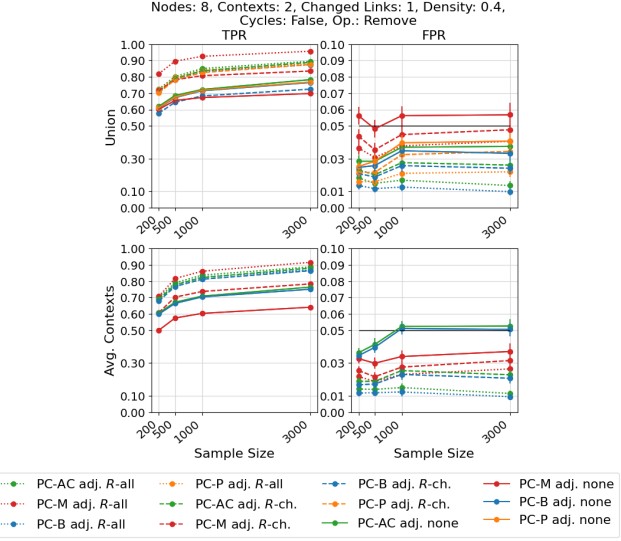

Figure 12: TPR and FPR results for the setup presented in Sec. 5.2 without cycles in the union graph, but with data imbalance factor $b = 1.5$. Here, we present results where all links to $R$ are known (adj.$R$-all), links to children of $R$ are known (adj. $R$-c.) or without any link assumptions (adj. none) for the methods PC-AC (our algorithm), PC-B (baseline), PC-M (masked data) and PC-P (pooled data). Results for the union graph are presented in the rows above, and results for the context-specific graphs averaged over contexts are presented in the rows below.

Results investigating the effect of data imbalance, presented in Fig. 12 show that our method, PC-AC, still performs better in terms of FPR or TPR compared to PC-M, despite a data imbalance factor of $b = 1.5$, if links from $R$ to its children are known or without any link assumptions, similar to the

results in the main paper. While the overall FPR increases slightly from the results presented in Fig. 3, especially for PC-P, this increase seems insignificant. We mention that, for this experiment, a total of 10 trials have failed (see Sec. E.2).

## E.4  Computational runtimes

We evaluate the computational runtimes of the different methods evaluated in Sec. 5.1 and App. E.3 and present these in Table 2. PC-AC has a slightly higher runtime than PC-M due to the additional tests involving the context variable. Among the methods that generate the union graph, the baseline PC-B has, as expected, the longest runtime, as it uses the information from both the PC-M results and the results of PC on the pooled data (PC-P). We also observe that CD-NOD seems to be the most time-efficient algorithm on pooled data, while PC is the most computational intensive. This difference in particular might arise from implementation details.

Table 2: Comparison of the computational runtimes in seconds of the PC-AC, PC-M, PC-B, PC, FCI and CD-NOD method that have been used in the experiments. While PC-AC and PC-M use only context-specific data, the other methods generate a union graph and / or use the pooled datasets, as discussed in Sec. 5.1. For the PC-AC and PC-M, we present the average time per regime in seconds.

| | **Context-specific data (avg. per regime)** | | **Union / pooled data** | | | |
| | **PC-AC** | **PC-M** | **PC** | **PC-B**
**PC(M + PC)** | **FCI** | **CD-NOD** |
|---|---|---|---|---|---|---|
| $n = 200$ | 2.784 | 2.345 | 1.419 | 4.203 | 0.874 | 0.775 |
| $n = 500$ | 3.254 | 2.679 | 1.693 | 4.947 | 1.054 | 0.974 |
| $n = 1000$ | 3.632 | 2.948 | 1.923 | 5.555 | 1.282 | 1.173 |
| $n = 2000$ | 4.444 | 3.579 | 2.382 | 6.826 | 1.636 | 1.451 |
| $n = 3000$ | 5.302 | 3.862 | 2.900 | 8.202 | 1.915 | 1.730 |

# F  Extensions and future work

In the Discussion Sec. 6, we have outlined a few ideas for extensions and future directions for the algorithm presented in this work. Here, we discuss these ideas in more detail.

## F.1  Orientation rules

In some cases, additional edge orientations beyond JCI arguments (15) are possible by using context-specific information. For instance, consider the following example:

**Example F.1.** Orientations Beyond JCI:
In context 0, we have $X \to Z \leftarrow Y$. In context 1, we have $X \to Y$, and no edges at $Z$.

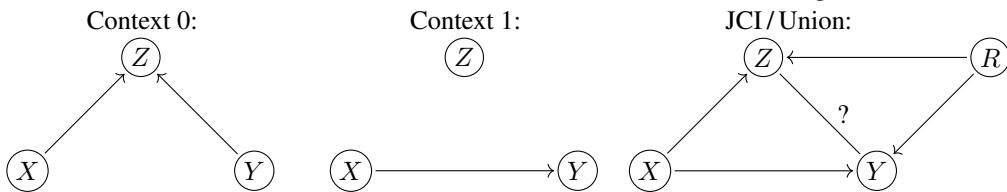

Context 0 can be oriented by the unshielded v-structure, context 1 can be oriented by a JCI argument (15). In the JCI case, both $X \to Z \leftarrow R$ and $X \to Y \leftarrow R$ are unshielded v-structures, thus leading to orientations. The v-structures $X \to Z \leftarrow Y$ and $R \to Z \leftarrow Y$ are both shielded. Furthermore, the v-structures that would occur for $Z \to Y$, i.e. $X \to Y \leftarrow Z$ and $R \to Y \leftarrow Z$ are both shielded and thus could not have been detected, and there is no conclusion that could be drawn from not detecting them. Thus the (pooled) independence structure is symmetric in $Y$ and $Z$. Therefore, also no graphical consistency criterion (avoid directed cycles etc.) can lead to an orientation of the edge between $Y$ and $Z$ and the graph obtained by JCI is also completely symmetric under exchange of $Y$ and $Z$, missing the orientation of the edge $Y \to Z$ (which in context 0 is found immediately by our method).

For the following example,

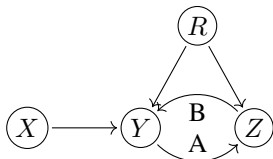

a slightly different problem occurs. Orientations are obtained by observing that $X \to Y \leftarrow R$ is unshielded, thus detectable, and in context A, $Y \leftarrow Z$ can be excluded, because it would introduce the unshielded v-structure $X \to Y \leftarrow Z$ (which is used for orientations in context B). This argument works only for the sparse union graph (as opposed to its acyclification) *and* only per context (the orientation is only well-defined per-context). The latter point makes it fundamentally inaccessible to JCI arguments, the former point also leads to the problem of Example F.1 (where it occurs independently of the second point).

## F.2  Deterministic indicators

In realistic examples, context indicators are often obtained by thresholding. An extension to such deterministic functions of observables seems possible as follows:

Given a deterministic indicator $C = g(W)$ for an observable $W$ ($g$ could be multi-variate, see below), add $C$ as a node to the output graph and draw a special edge $W \to C$ (indicating that this edge is deterministic and provided by expert knowledge). Then execute a standard constraint-based causal discovery method on the variables (excluding $C$) with these modifications for all $c$ values of $C$ occurring in data:

(i) If a test $X \perp\!\!\!\perp Y | Z$ is required, and $W \in Z$, then test $X \perp\!\!\!\perp Y | Z, g(W) = c$ instead.

(ii) If the resulting regime-specific skeleta differ from the union graph in the removal of an edge $X - Y$, then:

   (a) If both $X$ and $Y$ are adjacent to $W$, save the edge $(X, Y)$ as ambiguous for (iii).

   (b) If neither $X$ nor $Y$ are adjacent to $W$ (which could happen if the links to $W$ are deleted with larger adjustment sets), then mark the link as special (indicating that this is not a physical change by Lemma C.4).

   (c) If only $Y$ (or only $X$) is a child of $W$, draw a (directed) edge $C \to Y$ (or $C \to X$).

(iii) After the orientation phase (potentially using Sec. F.1), for each ambiguous edge $(X, Y)$ saved in (ii) (again using Lemma C.4):

   (a) If the edge was oriented $X \to Y$, draw a (directed) edge $C \to Y$.

   (b) If the edge was oriented $X \leftarrow Y$, draw a (directed) edge $C \to X$.

   (c) If the edge was not oriented, draw special (indicating to the user that they are ambiguous, possibly only one of them real) directed edges $C \to X$ and $C \to Y$.

In this scenario, testing $X \perp\!\!\!\perp W | g(W) = c$ could provide additional information:

**Example F.2.** Qualitative and Quantitative Context Dependence:
Assume $C = g(W)$ as above is binary, and

$$Y = \big(\alpha + \mathbb{1}(C)\big) \times (1 + \beta W) \times X + \eta_Y.$$

If $\alpha = 0$, then the mechanism depends *qualitatively* on $C$: If $C = 0$, then $Y$ does not depend on $X$. If $\beta \neq 0$, then the mechanism depends *quantitatively* on $W$.

In the qualitative case ($\alpha = 0$), the CSI changes: $X \perp\!\!\!\perp Y | C = 0$. In the quantitative case ($\beta \neq 0$), $Y \not\perp\!\!\!\perp Y | C$ (note that one could even check $Y \not\perp\!\!\!\perp Y | C = c$ for further info), while in the non-quantitative case ($\beta = 0$) $Y \perp\!\!\!\perp Y | C$.

