# OpenReview forum: "Causal discovery with endogenous context variables"
_NeurIPS.cc/2024/Conference — NeurIPS 2024 poster_

### Official Review · Reviewer_7t5U · 2024-06-18

**Soundness:** 2
**Presentation:** 2
**Contribution:** 2
**Rating:** 5
**Confidence:** 3

**Summary:**

The authors tackle the challenging causal discovery task, namely, causal discovery from the pooled dataset collected under different environments, where the environment (a.k.a., the context) can be dependent on the system (endogenous) variables (i.e., the variables whose causality we are interested in). The proposed algorithm is simple and intuitive: It employs a modified PC algorithm using context-dependent independence tests. Although I am not so familiar with this topic, overall, I have enjoyed reading the paper. However, there are several clarity issues, so I hope that my comments will be helpful for paper revision.

**Strengths:**

- Addressing the context that is dependent on system variables is an important problem.

- Theoretical results seem sound (although I am non-expert of this field and did not follow the details).

- Overall, the paper is well written (though there is much room for improvement).

**Weaknesses:**

Below I will enumerate several clarity issues.

Section 1

* While the first and the second paragraphs in Section 1 are very clear, the third one was a bit disappointing. The authors suddenly introduce several technical terms, such as context-specific models, single causal union graph, and cyclic union graph, etc, without definitions or explications. Readers cannot understand, for instance, which graph in Figure 1 is a « single causal union graph » because there is no description about it.

* « as exemplified in Example 3.1 » Example 3.1 is very far away from Introduction. Thus, this paragraph is not well structured.

Section 3

* Relationship between the notion of intervention and the context indicator is unclear. I am not expert of causal discovery from multiple datasets like JCI algorithm, but I understand that the context is a more general notion than intervention: Some interventions can be represented using context variables, but not vice versa. Am I correct? For instance, how can non-stationary time series be formulated using context variables? Please elaborate the context variables more (in Appendix, if there is not enough space).

* Related to the above, Example 3.1 can be regarded as soft intervention on $Y$?

* Section 3.1 involving Example 3.2 is very hard to follow, although the example is intuitive. Lines 141-149 are very difficult to follow. Please introduce each variable (e.g., ice cream sale) together with notation (e.g., $Y$). Notation is inconsistent: Sometimes $R=0$, and sometimes $R=« ice »$. This paragraph should be clearer to clarify the significance of this work.

* In Section 3.2, each graph is not clearly explained. For instance, is the mechanism graph « a causal graph in a standard sense »? If so, state it clearly.

* Many notations are introduced without definition. Examples include $f_i|_{support P(Pa_i)}$, $\mathcal{F}^M$, $P_M$, and $G_{R=r}$ (Section 4.1). This makes it hard to follow the paper. Please proofread the paper before submission.

Section 4

* I am not so familiar with the statistical test for context-specific independence, but could you elaborate the statistical procedure for the testing? Are you simply picking up only the subset of data instances with $R=r$? Is it statistically reliable? Is the test statistic identical with the usual independence testing?

**Questions:**

NA

---

> ### Author Rebuttal · Authors · 2024-08-07
>
> We thank the reviewer for their time and effort in reviewing our paper and for acknowledging the importance of the problem we are working on.
>
> We address the mentioned weaknesses as follows:
>
> -   **Section 1:** We agree that the third paragraph should be thoroughly reworked. It fails to make the point it is supposed to due to excessive jargon and details. We propose the following changes: 1) shorten the paragraph into two paragraphs, one discussing the challenges of only looking into context-specific independencies and one highlighting the contribution of our method. 2) remove jargon unless necessary, and explain any introduced jargon in more detail. 3) change Figure 1 as described in the response to reviewer 9Xqc, introducing the definitions of context-specific graphs and union graphs. We will also move Example 3.1, possibly to the introduction, or avoid referring to it, depending on what is best for the reading flow.
>
> -   **Section 3:** We agree with the reviewer that some confusion might arise from our definition of a context variable. In our case, the choice of context variable is primarily restricted by the regime-sufficiency assumptions. However, the context variable can be selected from or introduced into the dataset by expert knowledge or any regime detection algorithm, as discussed in the response to reviewers 9Xqc and J2yL. We focus specifically on understanding under which assumptions on the context variable we can obtain reliable results and how these results can be interpreted.
>
>     If we understand correctly, the reviewer is considering context variables to be something like Pearl's $\mathcal{F}$-variables. These $\mathcal{F}$-variables are functional variables (taking values in function spaces) that allow writing any (hard or soft) interventions implicitly: If $f_Y(x)$ is the mechanism at $x$, depending on parents $x$ (possibly multivariate), one can add a parent $\mathcal{F}_Y$ taking a function (e.g., the original $f_Y$) as a value and replace the mechanism at $Y$ by the evaluation map eval($f, x$) $:= f(x)$. Since $\mathcal{F}_Y$ can take a value for $f$ that represents an intervention (e.g., a constant function for a hard intervention), this generalizes interventions. Example 3.1 can indeed be regarded as a kind of soft intervention on $Y$.
>
>     Context variables are an orthogonal concept: Given multiple distinct datasets over the same set of variables, the context injects information about the specific associations in each dataset to the analysis by appending a pseudo-variable to the dataset. The value of this pseudo-variable typically is a dataset index. This context could be functional, but in general, both contexts and functional nodes coexist independently of each other.
>
>     We are interested in context-specific changes. A possible viewpoint on the strong regime-sufficiency assumption (cf. Example 3.1) is that the context variable behaves like a functional node that implements a soft intervention removing some of the parents' effects on certain variables. In comparison to JCI [17], our method reveals the internal structure of some functional nodes: Beyond discovering the children of the functional node, it additionally uncovers the graphical changes induced by the implemented soft intervention.
>
>     Generally, phenomena like drifts can be modeled using (possibly continuous) contexts but may not have a simple interventional interpretation. Temporal regimes were an important motivation for our study of context-specific behavior. To make the connection to non-stationary time series, these can be formulated using context variables. For the context to be categorical (which we and others assume), the non-stationarity cannot be a drift, etc., but must be driven by a regime change, i.e., there must be temporal regions sharing a model or context. By finding the contexts in which the time series is stationary and describing the causal relationships for those stationary parts of the time series, we can describe the non-stationary system as a multi-context system.
>
>     We will introduce these discussions and clarify the differences between other types of special variables, such as $\mathcal{F}$-variables, in the Appendix.
>
> -   **Section 4:** Our method combines testing on the pooled and context-specific data. Context-specific independence (CSI) is indeed simply tested on the samples satisfying $R=r$. Since $R$ is categorical and $P(R=r)>0$ by assumption, $P_r := P(X,Y,Z|R=r)$ is a well-defined distribution. Therefore, testing conditional independencies on $P_r$ is well-defined and no different from any other independence test (albeit assumptions like linearity need some thought, see §B.12). The difficulty lies in the interpretation of the result, as fixing $R$ can lead to selection bias. Overcoming this problem is one of the main contributions of our paper and method.
>
>     There is a difference between testing links involving $R$ and not involving $R$ ($R$ is categorical) as discussed in §B.12. Beyond this, a further difference between testing on the pooled and context-specific data is the number of samples for each test. This approach is as reliable as the finite sample properties of each test being used. Lastly, the more general problem of multiple testing, from which constraint-based causal discovery methods generally suffer, can also impact reliability. For this reason, we consider it advantageous that our approach reduces the number of tests.

---

> ### Comment · Reviewer_7t5U · 2024-08-10
> **Response**
>
> Thank you for your response.
>
> I roughly understand what the authors mean, but still, I cannot clearly understand the definition of context variables and related graphical concepts. Since they are strongly relevant to the study, their poor clarity will reduce the impact of this work. For this reason, I will maintain my overall rating, and I hope that the authors can clearly explain these notions more.

---

### Official Review · Reviewer_J2yL · 2024-07-07

**Soundness:** 3
**Presentation:** 2
**Contribution:** 2
**Rating:** 5
**Confidence:** 3

**Summary:**

This paper proposes a constraint-based algorithm for context-specific causal discovery, which accommodates endogenous context variables.

**Strengths:**

1. This paper is well-motivated. In particular, I agree that it is important to investigate the case where the context variable is endogenous.

2. This paper provides theoretical guarantee for their proposed method.

3. This paper provides extensive experimental results to demonstrate the effectiveness of the proposed method.

**Weaknesses:**

1. The proposed method assumes that the context variable is both observed and known. In real-world applications, we may not know which observed variable is the context variable. Furthermore, the context variable might be a latent variable.

2. This paper lacks readability. It entails too many definitions and notations. It might be better if the authors could summarize these things in Appendix for reference. Besides, the authors would better defer more details to appendix rather than present almost all results in the main text.

**Questions:**

In line 80, the authors claim that "where the context can also be endogenous (in [17])", but in line 89, the authors also claim that "all above-mentioned methods assume that the context variable is exogenous.", which is quite confusing. If [17] allows endogenous context variables, please detail the difference between [17] and this work.

**Limitations:**

Please see weaknesses.

---

> ### Author Rebuttal · Authors · 2024-08-07
>
> We thank the reviewer for reviewing our paper and for acknowledging the strengths of our work.
>
> We now address the weaknesses as follows:
>
> * **Context variable:** We agree with the reviewer that the assumptions we have made may not always reflect real-world use cases. However, as we have pointed out in the answer to reviewer 9Xqc, we believe that our work lays a first step towards a more general solution while considering important possible implications, such as the problems arising from shifting observational supports. Since regime detection algorithms exist, it is possible to discover $R$ rather than having a hard-coded specific choice. Our method is modular and can be combined with expert knowledge or any detection algorithm providing a context. In our work, we give assumptions to clearly define what would make a good choice for a context variable and consider that the detection part is out of scope for our work. We also believe that further extensions, such as to the case with latent context variables, are out of scope here, as we have detailed in the answer to reviewer 9Xqc. We wish for our work to further encourage the community to think about extensions.
>
> * **Readability:** We agree with the reviewer that our paper may still need improvement in terms of readability. While we have tried to reduce the notations and definitions in the main paper to a minimum, we will consider how to further improve this. Furthermore, we will introduce the changes that we have also mentioned in the answers for the other reviewers 9Xqc, DjMB, and 7t5U.
>
>
> * **Question:**  We agree with the reviewer that this might be confusing. What we meant is: In the most general form of the formulation of context-specific graphs, [17] allows the context variable to be endogenous, see eq (3) and JCI assumption 0 in [17]. The core difference can be seen, e.g., from eq. (3): [17] describes the distribution over the union of the context-specific graphs, whereas we obtain more (also context-specific) information compared to this by analyzing the context-specific graphs. In the end, [17] proposes an algorithm for causal discovery from the pooled data, while we go beyond this information to further obtain context-specific causal graphs, using both the pooled and the context-specific tests. We will add this discussion to the main paper, as we believe that it is important to more clearly differentiate our work from [17].

---

> > ### Comment · Reviewer_J2yL · 2024-08-08
> > **Response to authors**
> >
> > Thanks for your hard work. I have read your rebuttal and I'm now sure that I have no misunderstanding about this paper. I agree that this paper has provided some novel insights. But considering that this work is not the first one investigating the problem of causal discovery with endogenous context variables and this work remains at a relatively elementary stage (both detection and latent context variables are out of your scope), I decide to maintain my borderline accept score.

---

> > > ### Author Response · Authors · 2024-08-09
> > >
> > > We are sorry we could not change your mind. Nevertheless, we do not want to leave the new input unaddressed: After the introduction, we give a list of four main points where the paper makes novel contributions. Indeed one of those points is "[...] adapting existing constraint-based CD algorithms to extract context-specific information, in addition to joint information [...]". We are unsure which previous work the reviewer is referring to. Assuming this extends on the original question about [17], we want to clarify again that [17] is concerned with joint information and does not consider context-specific information. They employ the same setup to study a different problem.

---

> > > > ### Comment · Reviewer_J2yL · 2024-08-09
> > > >
> > > > I think the contribution of this work is somewhat limited because
> > > >
> > > > - This work is not the first one investigating causal discovery with context-specific variables. Some previous works such as [1] has also discussed this problem.
> > > >
> > > > - Both detection of context variables and latent context variables are out of your scope.
> > > >
> > > > [1] Context-specific causal discovery for categorical data using staged trees. AISTAT 2023.

---

> > > > > ### Author Response · Authors · 2024-08-13
> > > > > **Response to Comment by J2yL**
> > > > >
> > > > > We thank the reviewer for pointing out their reference [1], henceforth referred to as [ref1].
> > > > >
> > > > > To the best of our understanding, the referenced paper approaches the problem from a similar perspective as the work of reference [22] from our paper on LDAGs: The authors of [ref1] seem to be primarily interested in independence structures. The basic idea of the staged trees they use seems to be that for a purely categorical model independence $P(Y=y|X)=P(Y=y)$ simply means that \emph{all} conditionals $X=x$ lead to the same probability, $P(Y=y|X=x_i)=P(Y=y|X=x_j)$. Therefore, one can generalize independence by looking at the equalities of values in conditional independence tables (replace "all" by "some" in the previous sentence). This generalization includes CSI
> > > > > as a special case.
> > > > >
> > > > > While [ref1] has valuable contributions, there are a number points where the approach substantially differs from ours:
> > > > >
> > > > > * Their core discussion is on (generalized) independence structure.
> > > > > * Their approach is inherently tailored to purely categorical systems.
> > > > > * The relation to causality of the staged trees approach is indirect: In [ref1], SCMs are never discussed. Instead, the authors use a global causal order (which is assumed to be known) to obtain interventional distributions directly from a Markov factorization (a tree-like one along the total order, rather than the one along a graph customary with SCMs).
> > > > > * Their approach to finding this total causal order (via BIC) should not work generally (eg with continuous variables, see Prop. 4.1 (p.44) from Peters, Janzing, and Schölkopf. Elements of causal inference: foundations and learning algorithms.
> > > > > * They cannot use sparsity of the graph for efficiency, while our PC-like approach does.
> > > > > * They discuss neither causal (SCM-based) interpretations of the results nor the assumptions required to this end, while our work focuses specifically on those challenges.
> > > > > * They do not specify whether acyclicity is assumed, but the discussion around equation 4 seems to implicitly assume acyclicity.
> > > > >
> > > > >
> > > > > While we agree that there are similarities to our work, the approach of [ref1] seems less related to our work than LDAGs. This is why we we have discussed LDAGs in detail in the main paper. Furthermore, we would like to point out again that a core difficulty and insight within our work is the study of observational-support-related problems.
> > > > > We believe treatments focused on independence structures and typical (SCM-based) treatments of causality have very different underlying ideas, strengths and assumptions. To some degree, our results can bridge the gap between SCM based approaches and independence models, which we consider an important contribution of our work.

---

### Official Review · Reviewer_DjMB · 2024-07-13

**Soundness:** 3
**Presentation:** 2
**Contribution:** 3
**Rating:** 6
**Confidence:** 3

**Summary:**

The authors consider an SCM $M$ with a labelled and observed context variable $R$ that is endogenous, i.e., causally depends on other variables in the model. In this setting it is generally not true that $P_{M}(\ldots \mid R=r) = P_{M}(\ldots \mid \mathrm{do}(R=r))$. Therefore, selecting only data for which $R=r$ and applying constraint based causal discovery algorithms - as one might do with exogenous context variables - will introduce selection bias when estimating context-specific graphs $G^{\text{phys}}_{R=r} := G[M \mid \mathrm{do} (R=r)]$.

The authors propose an algorithm that adaptively selects whether to do a conditional independence test (CIT) using context specific or pooled data. Their method is provably complete (recovering the population graph of the underlying SCM with oracle CITs) under a set of assumptions that relate both to the underlying SCM and to the (context specific) observational distribution.

**Strengths:**

The authors motivate the challenges of efficient causal discovery with endogenous context variables clearly. They thoroughly document the challenges to their approach, especially in the appendix, and highlight potential avenues for future work.

While their submission needs refining, particularly on the clarity of their definitions, the authors provide structure for - and a novel solution to - an understudied causal discovery problem with many significant real-world applications.

**Weaknesses:**

(Presentation) The authors should present their results more carefully by using terms only once they are defined and being consistent with notation. For instance, the motivation behind a "$G^{\text{phys}}_{R=r}$" is considered informally in the discovery goals section (a few pages before the notation is used), but the notation itself is defined neither formally nor informally in the main text. The definition is left to appendix but is not cross-referenced in the main text. Similarly, I have no idea what the "AR" in their method "PC-AR" actually stands for.

The structure of the paper unfortunately makes simple typos (e.g., "$G^{\text{descr}}_{R=r} \subset G^{\text{union}} = G^{\text{union}}$" on line 262) difficult to interpret or correct by the reader. If the authors could restructure the paper to assist the reader in understanding their technical contributions this would greatly impact my evaluation of the submission's presentation. For instance, the physical and descriptive context graphs could have simply been defined in the discovery goals section.

(Interpretation of strong context-sufficiency) It is not clear to me what the authors mean by "injective" in the definition of strong context-sufficiency. I will assume that "$f_i$" refers to the structural equation for variable $X_i$ (the authors did not define this in the text) and take "injective" to literally mean $f_i$ has at most one input that corresponds to any output in $\mathrm{dom}(X_i)$. This seems to me to be a very restrictive assumption: say $f_i$ had one parent $A$ and a noise term $e$ and both were, e.g., normally distributed, wouldn't the "injective" property preclude any model like $f_i(A, e) = A + e$?

(Experimental design and results) In the experiments, cycle length in the union graph was not controlled despite this being required for the authors' method to recover $G^{\text{phys}}_{R=r}$. This complicates comparison with the brute-force baseline "PC-B". Furthermore, details on the time taken to run PC-B versus PC-AR should have been included to understand the computational advantage of the authors' approach.

**Questions:**

(Algorithm 1) There is no definition of the dummy variable $i$ in Algorithm 1. Is it meant to be looped over in line 4? Is line 12 in Algorithm 1 rather meant to say "remove $X_i - X_j$ from $G_{R=r}$"?

**Limitations:**

Yes - the authors make good effort to document the challenges to their approach and motivate potential future work.

---

> ### Author Rebuttal · Authors · 2024-08-06
>
> We thank the reviewer for their time and effort in reviewing our paper. We also thank the reviewer for acknowledging the strengths of our paper, and that we have made an effort to document all possible limitations of our approach. We address the weaknesses as follows:
>
> * **Notation and typos:** We realize that in some places our notations have not been introduced, which we will fix in the final version of the paper (main text and appendix). We will revise all notations to ensure that no notation comes without a proper introduction. We will also fix the typo to simply say $G^{\text{descr}}_{R=r} \subset G^{\text{union}}$ (which was intended in that place). There are some unexplained notations, such as what AR stands for ("adaptive regimes"). We will address these problems as well.
>
> * **Restructuring for clarity:** We will introduce a table better explaining the definitions of graphical objects better (see general post). We believe this will considerably help the reader to obtain an overview. We will also revise the Discovery goals sections and combine it with the Graphical objects section in order to improve the reading flow. Further, we will change the introduction such that it is easier to read, as described in the general post.
>
> 	The third paragraph of the introduction does not seem to make the point we actually wanted to illustrate, and instead contains too many details and jargon (as also pointed out by reviewer 7t5U). We will improve this paragraph. Generally, we will reduce jargon in both the abstract and introduction, and we will more clearly introduce required terms such as system and context variables.
>
> * **Strong context sufficiency:** Thank you for this remark! Indeed, this definition needs a revision. The following is what is supposed to happen when strong context-sufficiency is fulfilled: If the observational support given $R=r$ of, say, $X$, is restricted to a region where a mechanism $f_Y(X, \ldots)$ is constant in $X$ can make the link from $X$ to $Y$ disappear when doing independence testing.
>
> 	Excluding CSI arising like this requires a way of saying what "constant in a region" means. The single-graph sufficiency assumption formally captures this requirement and restricts it to the most intuitive class of models that do not suffer from the support issues mentioned in Example 3.2. Example 3.1 shows that there clearly is a (rather large) class of intuitive models which look like $f_Y(X,R,\ldots) = 1(R)g(X,\ldots) + h(\ldots)$. Now, one still has to exclude that $g$ or $h$ are constant in a region. The strong sufficiency condition aims to capture this intuitive notion, and one way of ensuring a map is *not* constant on any region is to require it to be injective: Then region simply means any set containing more than one point.
>
> 	An example of a model that *should* fulfill this assumption is a model which becomes linear after fixing $R$. However, as pointed out correctly by the reviewer, a linear map $\mathbb{R}^n \rightarrow \mathbb{R}$ can only be injective if $n\leq1$. This is not the case here, since every mechanism also depends on its noise term. This lack of applicability can be fixed by requiring injectivity "for each argument" (i.e. while holding all other parents at any fixed value, for example for an additive mechanism $f_Y(X_1, X_2, \ldots)=f^1_Y(X_1) + f^2_Y(X_2) + \ldots$ this means each $f_Y^i$ has to be injective). This actually includes all models which are linear after fixing, i.e., intervening to set, $R=r$. We will fix this and probably add the linear and general additive cases as examples to the appendix as well.
>
> 	The strong sufficiency assumption provides a class of examples for which we can obtain reliable outcomes with our method. However, there are many possible choices for the precise formal statement of a strong sufficiency assumption in this sense. For example, one could require, that the noise distributions are absolutely continuous and that the causal mechanisms are differentiable, and such that regions where any partial derivative is zero have measure zero. We will add a remark to the appendix discussing our choice of assumption more clearly.
>
> * **Simulation study:** We thank the reviewer for this remark, we agree that we should have specified this. From 71 samples, only two samples have a cycle length 3, while for the rest we only have cycles length 2. Therefore, we believe that the comparison is still informative. We will also include a plot with the comparison of the run times of all evaluated methods in the final version of the paper.
>
> * **Question to Algorithm 1:** There are indeed some typos in the algorithm. First $j$ is iterated, than $i$ is iterated over adjacencies of $j$ and $S$ over subsets of a specific size of adjacencies of $j$ excluding $i$. This will encounter all unordered pairs (thus edges) of $i, j$ twice (once with names/order exchanged), so searching for $S$ in adjacenices of only $j$ (but not of $i$, whose adjacencies will be searched when encountering the pair the second time, in reversed order) suffices. As the reviewer pointed out, line 12 should indeed say $G_{R=r}$, and we will change this as well.

---

> > ### Comment · Reviewer_DjMB · 2024-08-13
> >
> > Many thanks to the authors for their responses, their clarifications and their revisions.
> >
> > I find the authors' approach quite exciting, especially their insights into challenges for large cycles and unknown endogenous context variables. However, it is difficult to understand from rebuttals alone what impact their (quite heavily) revised presentation will have on the overall clarity of their work. For these reasons, I am willing to revise my score for the soundness of their approach to 3: Good, and my overall score to 6: Weak Accept.

---

### Official Review · Reviewer_9Xqc · 2024-07-14

**Soundness:** 3
**Presentation:** 2
**Contribution:** 2
**Rating:** 5
**Confidence:** 4

**Summary:**

The authors address the problem of causal discovery in situations where causal relationships change across different contexts. They propose a modified version of the PC algorithm, which either performs a conditional independence test (CIT) on pooled data or context-specific data, depending on the scenario. The paper presents several theoretical results for this modified algorithm and evaluates its performance through various empirical studies.

**Strengths:**

The authors tackle an important and relevant research problem, and their theoretical findings are noteworthy, especially the set of sufficient assumptions under which they derive their results. These assumptions could be valuable for future research in this area.

**Weaknesses:**

The organization of the paper makes it difficult to follow. For instance, numerous graphical definitions are presented without examples or figures, even in the appendix, which would aid understanding. I strongly suggest at least a running simple graphical example to show different graphs. Additionally, the definitions of the main assumptions are not included in the main text, further complicating comprehension.

The results rely on the assumption of causal sufficiency, but it is unclear how these results could be generalized to scenarios with hidden variables. Another significant limitation of the proposed framework, as acknowledged by the authors, is the requirement for knowledge of the context indicator.

The proposed algorithm require $Adj(X_j)$ and whether R belongs to this set for all $j$. This knowledge is obtain by a standard PC algorithm using pooled data. What happen if the output of this PC algorithm is noisy, e.g., it returns that R belongs to the conditioning set but in reality it is not?

While the proposed algorithm clearly outperforms the baselines in certain settings, the margin of improvement is relatively small, particularly when the conditional independence test is non-parametric.

**Questions:**

Please see above.

**Limitations:**

Please see above.

---

> ### Author Rebuttal · Authors · 2024-08-06
>
> We thank you for reviewing our paper, and for acknowledging the importance of the problem. Furthermore, we thank the reviewer for appreciating the value of the set of assumptions we have derived for a first step towards solving the problem of endogenous context variables.
> We address the weaknesses as follows:
>
> * **Graphical definitions:** We agree with the reviewer that our paper should be improved to aid understanding, we address this by the changes listed in the general post. In particular we give a table to systematize the graphical objects better.
>
> * **Main assumptions:** We decided to only informally introduce the assumptions, as we believed that technical detail might actually hinder the gaining of intuition. We will have another look at this paragraph.
>
> * **Causal sufficiency:** As we are the first to consider the implications of changing observational support for context-specific causal discovery with endogenous context variables, we started to tackle the problem with a simple setup. Hidden confounders are a problem very different from the one we study. It is common in the literature to initially assume causal sufficiency, albeit there are some doubts about its realism. Difficulties to a generalization with hidden confounders could arise on multiple levels:
> 	* One has to understand how the ideas underlying the graphical objects we are currently using must be adapted.
> 	* The Markov-property combines independence tests from the pooled data and context-specific data into a single graphical object. To this end, the standard d-separation argument is modified. One has to verify this approach still works when causal sufficiency does not hold.
> 	* One has to ensure that the right tests are executed in an efficient way, which should be possible, by replacing the PC algorithm, for example, by the FCI algorithm.
>
> * **Context indicator:** We agree with the reviewer that the the fact that we must know R in advance can be a limitation. However, we would not necessarily consider it a weakness. Regime detection algorithms can be used to discover R rather than having a hard coded specific choice. Our method is *modular* and can be combined with expert knowledge or any detection algorithm providing a context.
>
> * **Noisy PC output:** Typically, with the PC algorithm, errors in the early stages of the CI testing propagate, and our algorithm is subject to this flaw as well. It is helpful to discuss false negatives and false positives separately, because they lead to different downstream effects:
>
> 	False negatives (a link involving $R$ is not found): After the false negative, the algorithm behaves like standard causal discovery, because with $R$ removed from the adjacencies (thus from possible separating sets) no CSI is tested. It does not find additional context specific information, but it also does not incur further errors (beyond those from the causal discovery algorithm e.g. PC). So here finite-sample results can be interpreted in a spirit similar to how assumption-violations are understood in Rmk. 4.2: The result is still as correct as it would have been without our modification, but false negatives can lead to incomplete discovery of context-specific information.
>
> 	False positives (a link involving $R$ is detected to a non-adjacent $Y$): Assuming no simultaneous violation of regime-sufficiency assumptions, if our method discovers false positives, it behaves like the baseline (intersection-graph) method. Thus, in the presence of false positives, our method executes more tests than strictly necessary, because more adjacencies are found. Moreover, if R is erroneously found as adjacent, it will be added to the conditioning set, thereby running the tests for $R=r$, i.e., using the context-specific samples only. Conditioning on $R=r$ wrongly can induce selection bias and can also lead to testing on a reduced sample size, leading to further finite sample effects. Therefore, the subsequent error-rate should be slightly higher than for standard causal discovery.
>
> * **Marginal improvement over baseline:** As we are not sure which baseline the reviewer refers to, we discuss:
>
> 	(a) Comparison including context-specific information (here *non-*parametric tests actually performed better, but we believe to understand this problem, see §B.11): Indeed, the improvement may seem small with the parametric test. However, using the non-parametric test shown in Figure 2, the CIT is seen to significantly impact performance: We see much larger differences, especially in FPR, when using a non-parametric CIT compared to masking, testing on the pooled data or the (intersection-graph) baseline. Our baseline here is the intersection graph (§B.10), which to our knowledge has not been theoretically studied before. Technically, one of the advantages against this baseline is scalability (§B.9), even though in practice one would need a sufficiently stable causal discovery method, and this is not always the case.
>
> 	(b) Comparison to existing (pooled) methods: Compared to algorithms not using CSI, we provide additional information in the form of context-specific causal graphs.

---

### Author Rebuttal · Authors · 2024-08-07

## General Response to the Reviewers

We thank all reviewers for their valuable comments, and we are happy that they acknowledge the importance of our work and the value of our results. We agree with the reviewers that the presentation of the paper can be improved, and we summarize the main changes we aim to add to the final version of the paper as follows. Further questions are answered in the individual rebuttals.

* **Introduction:** We will improve the introduction by modifying Figure 1 to a figure which clearly explains the setup where a context variable is endogeneous, and will move the discussion around the current Figure 1 to Section 4. We will then move Example 3.1. to the introduction to better illustrate our goals. We will considerably improve the third paragraph of the introduction by removing unnecessary details and jargon, and rather explain better how our work relates to methods describing context-specific independencies (CSI), and how our assumptions allow us to give an interpretation in terms of structural causal models (SCMs) to the CSIs.


* **Graphical objects:** We agree that the graphical objects may be hard to understand at first sight in the current form of the paper. We therefore propose to add the following table (formatted as best as possible) which underlines the differences and connections between the different graphical objects:

	|  | Observable... | ...graphs... | ...$G[M,P]$ |
	| :----------------: | :------: | :----: | :----: |
	| Symbol |   $G^{\\text{descr}}_{R=r}$ |  $G^{\\text{phys}}_{R=r}$ |  $G^{\\text{union}}$ |
	| Name |   descriptive | physical | union ('standard') |
	| Model $M$    |  $M$ or $M_{\\text{do}}$ | $M_{\\text{do}}$ | $M$ |
	| Observational Support $P$ |  $P_M(...,R=r)$ | $P_M(...)$ | $P_M(...)$ |
	| Captured Information | independence-structure | altered mechanisms | union mechanisms |
	| Context-Specific ($r$-dep.) | yes | yes | no |
	| Used here primarily for | discovery | proofs | relation to literature |
	| Node Sets | system $\\cup \\{R\\}$ | system $\\cup \\{R\\}$ | system $\\cup \\{R\\}$ |
	| Edge Sets | active in context $r$ $\\subset$ | present in context $r$ $\\subset$ | in any context |

	We will also combine the graphical objects and the discovery goals sections for better reading flow.

* **Overall readability:** We will revise notation and try to further simplify it. We will better explain the assumptions. However, we believe that the exact definitions are better left to the Appendix. We will fix all discovered typos, including the typos in Algorithm 1.

* **Context variable:** We will revise our definition of a context variable to make it clearer that our method is modular and can be combined with any anomaly or regime-detection method. This change also makes it easier to explain how the current setup, where the context variable is measured and known, is plausible also for real-world scenarios.

## Redraft of third paragraph of introduction
The third paragraph of the introduction did not illustrate well what we actually wanted to say, which is (the following is an early redraft and has to be fit together with the remainder of the introduction, so is likely subject to change):

Multiple context-specific graphs can contain more qualitative
information than a single union graph as illustrated below:

**Example 3.1:** Given a binary context indicator variable $R$ and a *multivariate* mechanism of the form $ f_Y(X,R,\eta_Y) = \mathbb{1}(R) g(X) + \eta_Y,$ the dependence $X \rightarrow Y$ is present in the context $R=1$, but absent for $R=0$. This entails a context-specific independence (CSI) $X \perp Y | R=0$.

As indicated in the example, such additional information can be captured via context-specific independence (CSI). Graphical independence models describing the CSI structure of a dataset, for example, LDAGs, have been studied before [22]. However, a causal analysis, i.e., understanding interventional properties of the context-specific model, requires knowledge about the causal model properties.
In the single-context case, under the faithfulness assumption, knowledge about the causal properties of models is directly connected to the independence structure. As will be explored in detail in §3.1, this simple model to independence correspondence cannot generally hold in the multi-context case. Thus, an important open problem for the causal analysis of multi-context systems is the connection of CSI structure to the underlying causal model.

We provide a connection between the CSI structure and the underlying causal model and specify assumptions under which an efficiently computable subset of CSI and independencies on the pooled data together can be given an interpretation in terms of structural causal model (SCM) properties. The obtained context-specific graphs are of interest due to several desirable properties. For instance, context-specific modeling can avoid spurious edges that arise from cycles in the union graph, as shown in Figure 1.
Furthermore, CSI testing poses multiple finite-sample challenges: It dramatically increases the search space of independence testing. Additionally, CSIs are only tested on a subset of samples, thereby increasing the per-test error rate. Our approach, which requires the framework connecting CSI to causal properties, adaptively decides whether a specific test can be run on the pooled data, or must run on a subset of samples associated with a specific context. It executes only one of both tests and uses as many samples as possible, thereby improving finite-sample performance.

---

### Comment · Area_Chair_rcik · 2024-08-10

Dear Reviewers,

The authors have submitted the point-to-point response. Currently, the reviews are all around borderline accept/reject.
Could you all read the rebuttal and see if it addresses the main issues raised by your review?  (Thanks J2yL for the engagement.)
Do your ratings change? Are there still some unresolved issues?

AC

---

### Comment · Area_Chair_rcik · 2024-08-12

Dear Reviewers,

Thank you to those who engaged in the discussion with the authors.

Please confirm that you have read the authors' rebuttal and outline any concerns that remain (`9Xqc`, `DjMB`). Are there aspects of the rebuttal where the explanation is insufficient? If so, please specify. Additionally, please update your score to reflect your current view of the paper, even though you can change it later. This will give the authors a better sense of how your perspective has evolved.

Best,
AC

---

### Decision · Program_Chairs · 2024-09-25

**Decision:**

Accept (poster)

**Comment:**

The paper addresses the challenge of causal discovery in scenarios where causal relationships vary across different contexts, particularly when the context variable is endogenous, meaning it depends on other variables within the system. The authors propose a modified version of the PC algorithm that adaptively chooses between performing conditional independence tests on either pooled data or context-specific data, depending on the context. This approach is designed to avoid the selection bias that can occur when using traditional methods in the presence of endogenous context variables. The proposed algorithm is theoretically grounded, with proofs of its completeness under certain assumptions, and its effectiveness is demonstrated through empirical studies.

The reviewers are unanimously in favor of accepting the paper, though their support is somewhat tentative. There are several issues with clarity (definitions, assumptions, introduction, etc.), which I believe the authors can address during the revision. However, I am also a bit concerned about whether it may require another round of review, as it could involve a nontrivial amount of changes. One of several concerns is about the known context indicator. Given that this paper is the first to address an endogenous context variable, it is reasonable to assume a known context indicator. In the rebuttal, the authors claimed that a regime detection algorithm can be used to identify a context indicator, but no reference or clear explanation is provided on how this could be applied. If it is feasible, adding experiments with a learned context indicator would significantly strengthen the paper.

Given that the paper is well-motivated and the contribution is novel enough, I lean towards recommending the acceptance of the paper. The authors are strongly encouraged to revise the paper for better readability and also to add a discussion on learning a context indicator.